

# Sub-basin scale sea level budgets from satellite altimetry, Argo floats and satellite gravimetry in the North Atlantic

Marcel Kleinherenbrink[1], Riccardo Riva[1], and Yu Sun[1]

[1]Department of Geoscience and Remote Sensing, Delft University of Technology, P.O. Box 5048, 2600 GA Delft, The Netherlands

*Correspondence to:* Marcel Kleinherenbrink (m.kleinherenbrink@tudelft.nl)

**Abstract.** In this study for the first time an attempt is made to close the sea level budget on a sub-basin scale in terms of trend, annual amplitude and residual time series, after removing the trend, the semi-annual and annual signals. To obtain errors for altimetry and Argo full variance-covariance matrices are computed using correlation functions and their errors are fully propagated. For altimetry we apply a geographically dependent intermission bias (Ablain et al., 2015), which leads to differences in trends up to 0.8 mm yr$^{-1}$. Since Argo float measurements are non-homogeneously spaced, steric sea levels are first objectively interpolated onto a grid before averaging. For the Gravity Recovery And Climate Experiment (GRACE) gravity fields full variance-covariance matrices are used to propagate errors and statistically filter the gravity fields. We use four different filtered gravity field solutions and determine which post-processing strategy is best for budget closure. As a reference the standard 96-degree DDK5-filtered CSR solution is used to compute OBP. A comparison is made with two anistropic Wiener-filtered CSR solutions up to d/o 60 and 96 and a Wiener-filtered 90-degree ITSG solution. Budgets are computed for ten polygons in the North Atlantic, defined in a way that the error on the trend of Ocean Bottom Pressure (OBP) + steric sea level remains within 1 mm yr$^{-1}$. Using the anisotropic Wiener filter on CSR gravity fields expanded up to spherical harmonic degree 96, it is possible to close the sea level budget in nine-out-of-ten sub-basins in terms of trend. Wiener-filtered ITSG and the standard DDK5-filtered CSR solutions also close the trend budget, if a Glacial Isostatic Adjustment (GIA) correction error of 10-20 % is applied, however the performance of the DDK5-filtered solution strongly depends on the orientation of the polygon due to residual striping. In seven-out-of-ten sub-basins the budget of the annual cycle is closed, using the DDK5-filtered CSR or the Wiener-filtered ITSG solutions. The Wiener-filtered 60- and 96-degree CSR solution in combination with Argo lack amplitude and suffer from what appears to be hydrological leakage in the Amazon and Sahel regions. After reducing the trend, semi-annual and annual signals, 24-53 % of the residual variance in altimetry-derived sea level time series is explained by the combination of Argo steric sea levels and Wiener-filtered ITSG OBP. Based on this, we believe that the best overall solution for the OBP component of the sub-basin scale budgets is the Wiener-filtered ITSG gravity fields. The interannual variability is primarily a steric signal in the North Atlantic, so for this the choice of filter and gravity field solution is not really significant.



# 1 Introduction

Several studies attempted to close the sea level budget by using satellite altimetry, satellite gravimetry and observations or reanalyses of ocean temperature and salinity on a global scale. Closure of the budgets is required to get a consistent division between mass or Ocean Bottom Pressure (OBP) and steric related sea level changes. This helps us to identify the contributors to present day sea level changes. Contributors that affect ocean OBP are glacier and ice sheet melt and land water storage, while heat fluxes between ocean and atmosphere contribute to steric changes. Ocean dynamics have an effect on both the OBP and the steric change in sea level.

One of the first attempts to close the sea level budget compared time series of total sea level from satellite altimetry with the sum of the OBP component from satellite gravimetry and the steric component from Argo floats (Willis et al., 2008). This study showed that between 2003 and 2007 the sum and the total sea level have comparable seasonal and interannual sea level variability, however the 4-year trends did not agree. In that same year Cazenave et al. (2008) found comparable estimates of steric sea level estimated from Argo and from the difference between altimetry and the Gravity Recovery And Climate Experiment (GRACE) observations over 2003-2008. Using the same methods as Willis et al. (2008) the global sea level budget was closed within error by Leuliette and Miller (2009) and Leuliette and Willis (2011) over the period 2004-2008. All of the beforementioned studies used a form of reduced space objective interpolation (Bretherton et al., 1976) to create grids of Argo data. Li et al. (2013) attempted to close the global budget using temperature and salinity grids from Ishii et al. (2006).

While time series of satellite gravimetry and Argo observations became longer and the processing of gravity fields improved, it became possible to look at basin-scale budgets and patterns. Chambers and Willis (2010) compared gravimetry derived maps of Ocean Bottom Pressure (OBP) to those obtained with steric-corrected altimetry, whereas Marcos et al. (2011) investigated the distribution of steric and OBP contributions to sea level changes and looked at basin-scale differences. Purkey (2014) analysed differences between basin-scale OBP from satellite gravimetry and steric corrected altimetry using Conductivity-Temperature-Depth (CTD) profiles over the period 1992-2013. They showed that both methods captured the large-scale OBP change patterns, but that differences occur when deep-steric contributions below 1000 m are not considered. Over the North Atlantic Ocean the OBP trends were found to be statistically equal, but with large error bars for the steric-corrected altimetry results. Von Schuckmann et al. (2014) found global and large-scale regional (a third of the total ocean) consistency in sea level trends of the three systems in the Tropics as long as areas like the Tropical Asian Archipelago are not considered (Von Schuckmann et al., 2014); but they did not manage to close the budget between 30-60 degrees North and argued that the unability of Argo to resolve eddies in the western intensifications caused the difference in trends.

Some other studies focussed on sea level budgets in small basins. García et al. (2006) compared sea level trends in the Mediterranean from satellite altimetry, satellite gravimetry and the Estimating the Circulation and Climate of the Ocean (ECCO) model. ECCO is also used by Feng et al. (2012) to determine trends in the South China Sea. Time series of sea level budgets have been investigated in the Red Sea using Ishii grids (Feng et al., 2014).

Compared to previous studies, we improve the treatment of each dataset, in particular with respect to an accurate description of the uncertainties. We avoid using precomputed grids for Argo or altimetry and we use full variance-covariance matrices of the



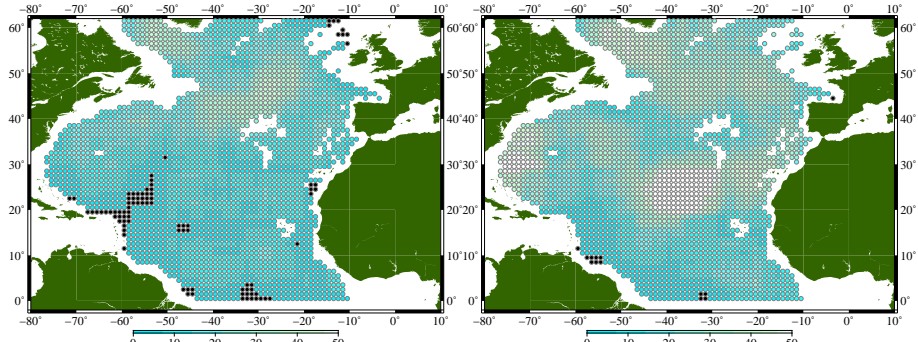

**Figure 1.** Number of Argo floats within a $10 \times 10$ degree box for grid cells where the depth is larger than 1000 m. Left: July 2004, right: July 2013. Only floats considered in this study are used for the statistics (Sect. 3.2). The black dots indicate no floats in the $10 \times 10$ degree box.

GRACE gravity field solutions. Secondly, we address the effect of several processing steps of particularly on gravimetry data in terms of trend, annual amplitude and (residual) time series. For altimetry we briefly discuss the effect of different averaging methods and analyse the effect on the trends of having a latitude dependent intermission bias (Ablain et al., 2015). For GRACE, DDK5-filtered solutions (Kusche, 2007)(Kusche et al., 2009) are compared with the anisotropic Wiener-filtered Klees et al.

(2008) solutions. Finally, basin and sub-basin scale budgets are created, problematic areas are identified and potential causes for non-closure are discussed.

This article will describe briefly the data used in Sect. 2. Secondly, the processing of the three datasets is discussed in the methodology section. In Sect. 4 the processed datasets are compared to existing products. The resulting basin and sub-basin scale budgets are described in Sect. 5. In the final section conclusions are drawn based on the results.

**2   Data description**

This section shortly discusses the data from the three observing systems that are used to determine the sea level budgets.

For the determination of the sum of the steric and the mass components of sea level satellite altimetry data are used. The altimetry data are obtained from the Radar Altimetry Database System (RADS) (Scharroo et al., 2012). RADS contains 1 Hz along-track data, which corresponds to an along-track separation of sea level measurements of approximately 6 km. The files

contain ranges, orbits and geophysical corrections for all altimeters that have been flown. In this study only the data of the Jason-1 and Jason-2 satellites are considered to have consistent spatial and temporal sampling over the period 2004-2014. The data of Jason-1 during its geodetic mission phase (2008-2013) are not used for the altimetry time series. Both satellites have a repeat-track of approximately ten days and the same orbital plane, which results in a ground-track separation of approximately 315 km, or 2.8 degrees, at the equator.



The steric component of sea level rise is determined using measurement profiles of temperature and salinity from the Argo float network. Since the first deployments of Argo floats in the year 1999, the number of Argo floats rapidly increased until approximately 3900 floats currently. Argo reached maturity around the year 2007, when at least 3000 floats were in the water (Canabes et al., 2013), which means that there is on average approximately one float per $3 \times 3$ degree box. For the North

Atlantic, steric sea levels can be analysed from 2004, because most areas are sampled already by Argo floats as shown in Fig. 1. However, in large parts south of the equator barely any float resurfaces, which will lead to problems in interpolation. In the North Atlantic the areas around the Antilles and north of Ireland are the only problematic areas. Most floats descend to a depth around 1000-2000 m and measure temperature and salinity while travelling upward. The resurfacing time of an Argo float is approximately 10-12 days. Using the distribution of temperature and salinity over depth the steric sea level is computed.

The Earth's time variable gravity field is measured since 2002 by the Gravity Recovery And Climate Experiment (GRACE). This mission measures changes in the Earth's gravity field by low Earth orbit satellite-to-satellite tracking. Traditionally the Earth's gravity field is expressed in spherical harmonics. In this study the release 5 monthly spherical harmonic solutions computed at CSR (Tapley et al., 2004) up to degree and order 60 and 96 are used, together with the ITSG-GRACE2016 solutions (Klinger et al., 2016) up to degree and order 90. All three products are provided with full variance-covariance or

normal matrices, which allows for statistically optimal filtering. In case of a proper error description, we expect that the results of the CSR 60- and 96-degree solutions give comparable results, except in areas with large gradients in gravity. However, since the variance-covariance matrices are comparable over the periods July 2003-December 2010 and February 2011-July 2013, but the orbit geometry substantially varies, the provided variance-covariance matrices are not expected to give a proper representation of the error, which leads to reduced quality filtering. As a consequence, the months July-October 2004 are

excluded from the analysis, when GRACE entered a near 4-day repeat-orbit. The addition of the ITSG solutions, enables us to compare an independent solution computed with a different approach to the standard CSR products. The non-dimensional gravity field coefficients are converted to units of Equivalent Water Height (EWH) before filtering, to make them compatible with the other two observing systems. For comparison, we also used the publicly available DDK-filtered solutions of CSR, however no variance-covariance matrices for those solutions are publicly available. In the processing phase, the Atmosperic

and Ocean De-aliasing Level-1B (AOD1B) product is incorporated (Dobslaw et al., 2013), which is based on the Ocean Model for Circulation and Tides (OMCT) and the European Centre for Medium-range Weather Forcecast (ECMWF) model. Monthly averages of the OMCT and the ECMWF are restored after processing to the time-varying gravity field in the form of spherical harmonics (Chambers and Willis, 2010), details on this are described in Sect. 3.3.

## 3 Methodology

The data described in the previous section are processed such that they are suited for establishing monthly regional sea level budgets. It implies that the equation

$$\bar{h}_{sla} = \bar{h}_{ssla} + \bar{h}_{obp} \tag{1}$$





**Figure 2.** List of geophysical correction applied in this study and the MSLs of NOAA.

|  | This study | NOAA |
| --- | --- | --- |
| Ionosphere | Smoothed dual-frequency | Smoothed dual-frequency |
| Wet troposphere | Radiometer | Radiometer |
| Dry troposphere | ECMWF | ECMWF |
| Ocean tide | GOT4.10 | GOT4.8 |
| Loading tide | GOT4.10 | GOT4.8 |
| Pole tide | Wahr | Wahr |
| Solid Earth tide | Cartwright | Cartwright |
| Sea state bias | Tran2012 | CLS11 |
| Dynamic atmosphere | MOG2D | MOG2D |

is satisfied within uncertainties, where $\bar{h}_{sla}$ is the Mean Sea Level (MSL) anomaly derived from the Jason satellites, $\bar{h}_{ssla}$ the mean steric sea level anomaly derived from Argo and $\bar{h}_{obp}$ the mean OBP in terms of EWH derived from GRACE. This section describes therefore the processing strategies for the three observation types from individual measurements to an average over a specified region in the ocean including the propagation of the formal errors.

As far as altimetry is concerned, after computing individual along-track sea level anomalies, two important processing steps are described in this section: a suitable averaging method to come to a time series of MSL for a given area and a way to deal with geographical dependencies of the intermission bias between the two Jason missions (Ablain et al., 2015).

To compute steric sea levels from Argo temperature and salinity measurements the TEOS-10 software is used (Pawlowicz et al., 2012). Since the Argo measurements are non-homogeneously distributed over the ocean, the steric sea levels are first

interpolated using an objective mapping procedure to a grid of $1 \times 1$ degree, before being averaged.

Monthly GRACE solutions of CSR and ITSG are provided with calibrated variance-covariance matrices, which allows the use of an anisotropic Wiener filter (Klees et al., 2008). Compared to other existing filters, it strongly reduces the stripes that are still present in the DDK-filtered solutions (as will be shown in Sect. 4, while not reducing the spatial resolution by applying a large Gaussian filter. A fan filter is applied after the optimal filter to reduce ringing artefacts that occur close to Greenland due

to the limited number of spherical harmonic coefficients (d/o 60-96).

## 3.1    Jason sea level

Individual sea level anomalies $h_{sla}$ measured with the Jason-1 and Jason-2 satellites are computed with respect to the mean sea surface ($mss$) DTU13 as:

$$h_{sla} = a - (R - \Delta R_{corr}) - mss, \tag{2}$$





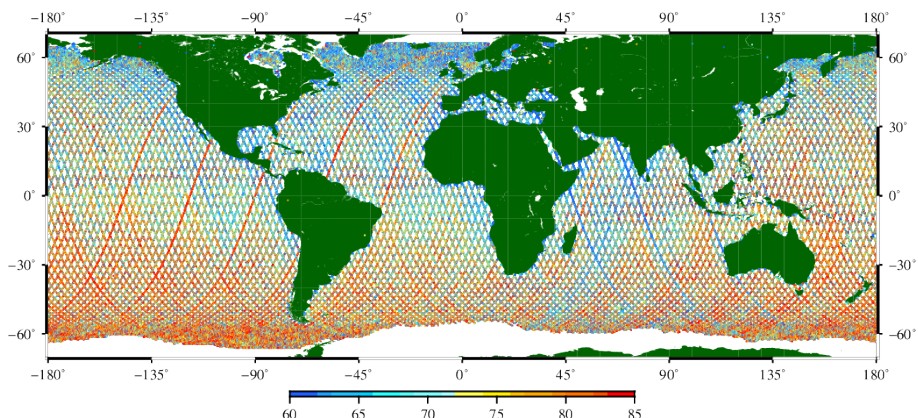

**Figure 3.** Geographical differences (mm) between Jason-1 and Jason-2 sea level estimates averaged over the tandem period.

where $a$ is the satellite altitude, $R$ the Ku-band range and $\Delta R_{corr}$ the applied geophysical corrections. The satellite altitude is taken from the GDR-D orbits. The latest versions of the geophysical correction are applied, as listed in Table 2. Sea level anomalies larger than 1 meter are removed from further processing, as in the NOAA GMSL time series (Masters et al., 2012).

In GMSL time series an intermission bias correction is applied, which is determined from the average GMSL difference

between Jason-1 and Jason-2 during their tandem phase, in which the satellites orbit the same plane only a minute apart (Nerem et al., 2010). However, the differences reveal a geographical dependence as shown in Fig. 3. Regional sea level budgets established in this study are proner to these geographical differences than when estimating global sea level budgets. This problem is partly corrected for by estimating a polynomial through the intermission differences, which only depends on latitude (Ablain et al., 2015) and is given by:

$$\Delta h_{sla,ib}(\lambda) = c_0 + c_1 \cdot \lambda + c_2 \cdot \lambda^2 + c_3 \cdot \lambda^3 + c_4 \cdot \lambda^4, \tag{3}$$

where $\lambda$ is the latitude and $\Delta h_{sla,ib}(\lambda)$ is the intermission correction. The sea level anomaly $h_{sla,c}$ corrected for intermission differences is then computed as:

$$h_{sla,c} = h_{sla} - \Delta h_{sla,ib}. \tag{4}$$

This correction is only applied to Jason-2 sea level anomalies. The parameters $c_n$, with $n = 0, 1.., 4$, depend on the applied

geosphysical corrections. For the corrections given in Table 2 the values for the parameters are given in Table 1. At halfway the North Atlantic, the intermission difference is several millimeters less than when only including the constant $c_0$ parameter (which slightly different if the other parameters are not estimated). This results in an approximate trend difference of several tenths of a millimeter over a period of 10 years.

Due to the limited sampling of the Argo network and the relatively large errors in the gravity field solutions it is necessary

to integrate sea level anomalies over extended areas. Previous studies producing GMSL time series have used two different techniques (Masters et al., 2012): gridding or latitude weighting based on the inclination of the orbit (Nerem, 1995). From





**Table 1.** Values for the parameters of the intermission difference correction.

| Parameter | Value | Unit |
|---|---|---|
| $c_0$ | 71.9 | mm |
| $c_1$ | $-74.7 \cdot 10^{-3}$ | mm deg$^{-1}$ |
| $c_2$ | $51.1 \cdot 10^{-5}$ | mm deg$^{-2}$ |
| $c_3$ | $-43.3 \cdot 10^{-7}$ | mm deg$^{-3}$ |
| $c_4$ | $-15.1 \cdot 10^{-8}$ | mm deg$^{-4}$ |

here on the latter is referred to as the 'Nerem method'. The gridding method is problematic when using the Jason satellites, because of their large track spacing at the Equator, causing the number of invidual observations per grid cell to decrease at low latitudes (Henry et al., 2014). A solution is to increase the grid cell size, but this has a disadvantage if sea level budgets are constructed over an irregular and/or a small polygon. The latitude weighting method has the disadvantage that it underweights measurements at high latitudes ($> 50$ degree) (Scharroo, 2006), because it assumes the number of measurements to go to infinity at the inclination of the satellites.

Therefore it is suggested to average the sea level anomalies based on the number of available measurements within a latitude band. The method connects the weights assigned to the measurements to the number of measurements $N_l$ in a latitude band $l$ of one degree and the area of the sea surface $A_l$ in the following way:

$$\omega_i(l) = \frac{A_l}{N_l}. \tag{5}$$

These weights are normalized:

$$w_i = \frac{\omega_i}{\sum_{i=1}^{I} \omega_i}. \tag{6}$$

A MSL anomaly $\bar{h}_{sla}$ for an area is computed with:

$$\bar{h}_{sla} = \hat{w}^T \hat{h}_{sla,c}, \tag{7}$$

where $\hat{w}$ is the vector of normalized weights and $\hat{h}_{sla,c}$ is the vector of sea level anomalies corrected for intermission differences.

For the error estimation variance-covariance matrices are computed as described in Le Traon et al. (1998). This method separates the long-wavelength errors from the representativity errors due to ocean variability. White measurement noise is not considered here, because it becomes very small when averaged over large areas. Among the long-wavelength errors, we consider the orbit, ocean tide and inverse barometer errors. These errors are assumed to be fully correlated between measurements within the track and uncorrelated between inter-track measurements. It is noted that those correlations do not hold over large basins ($> 2000$ km in the zonal direction) (Le Traon et al., 1998) and therefore the error is overestimated. For the other error sources an e-folding decorrelation time of 15 days is assumed and the zero crossing of the correlation distance function $d_{corr}$



is given by (Le Traon et al., 2001):

$$d_{corr} = 50 + 250 \frac{900}{\lambda_{avg}^2 + 900}, \tag{8}$$

where $\lambda_{avg}$ is the average latitude of two measurements. Ultimately, this results in equations for the covariance of respectively measurements in different tracks and on the same track:

$$\langle \epsilon_i, \epsilon_j \rangle = \rho_{ij} \sigma_{ov}^2$$

$$\langle \epsilon_i, \epsilon_j \rangle = \rho_{ij} \sigma_{ov}^2 + \sigma_{lw}^2, \tag{9}$$

where $\rho_{ij}$ is correlation computed with the decorrelation time and distance provided above, $\sigma_{ov}^2$ is the ocean variability variance and $\sigma_{lw}^2$ is the long-wavelength variance. The values for $\sigma_{ov}$ and $\sigma_{lw}$ are assumed 100 mm and 15 mm, where the first number comes from typical mesoscale variability (Chelton et al., 2007). By putting these equations in the variance-covariance matrix $C_{sla}$, the standard error $\bar{\sigma}_{sla}$ for the mean sea level anomaly is computed using:

$$\bar{\sigma}_{sla} = \sqrt{\hat{w} C_{sla} \hat{w}^T}. \tag{10}$$

Both the satellite altimetry mean sea level anomalies as well as the EWHs are affected by Glacial Isostatic Adjustment (GIA). In case of altimetric measurements this is corrected by subtracting the geoid trend from an Earth model with VM5a viscosity profile and ICE-6G deglaciation history (Peltier et al., 2015). Errors in the GIA trends are typically assumed to be in the order of 30% of the signal (Von Schuckmann et al., 2014).

Because the CSR gravity fields are created on a monthly basis and the altimetry measurements are averaged over a cycle of approximately ten days, the altimetry measurements are low-pass filtered. A low-pass filter $f_{lp}$ is computed by taking an Inverse Discrete Time Fourier Transform, which results in:

$$f_{lp} = \frac{\sin(2\pi f_c(t - t_m))}{\pi(t - t_m)}, \tag{11}$$

with $f_c$ the cut-off frequency, which is taken as 12 cyc/year, $t$ is time vector of the altimetry time series and $t_m$ the time at the middle of a month. This filter is infinitely long, so therefore we cut it at two months. To obtain a better frequency response the filter is windowed using a Hamming window $w_H$, so that:

$$w_H = 0.54 - 0.46 \cos(\frac{2\pi(t - t_m - L/2)}{L}), \tag{12}$$

where $L$ is the length of the window which is two months. The applied filter $h_{lp}$ is then written in the time domain as:

$$h_{lp} = f_{lp} \cdot w_H. \tag{13}$$

## 3.2 Argo steric sea level

From the individual Argo T/S-profiles steric sea levels are computed using the TEOS-10 package. This package requires the conversion of the PSS-78 practical salinity values measured by Argo to the absolute salinity $S_A$ (Grosso et al., 2010) as well



as the ITS-90 temperatures to conservative temperature $\Theta$. The TEOS-10 program numerically integrates the equation for the geostropic steric sea level (IAPSO, 2010):

$$h_{ssl} = -\frac{1}{g_0}\int_{P_0}^{P}\hat{\delta}(S_A(P'),\Theta(P'),P')dP',\tag{14}$$

where $P_0$ is the surface pressure, $P$ is the reference pressure, which is set to 1000 dBar (approximately 1000 meters depth) and $g_0$ is a constant gravitational acceleration of 9.763 m/s$^2$.

In the analysis only profiles that reach at least 1000 meters depth are included and at least have a measurement above 30 meters depth, which is the typical depth of the mixed layer. A 'virtual measurement' is created at 1 meter depth, assuming the same salinity and potential temperature values as the highest real measurement, so that the top steric signal is not missed. Only measurements that have error flag '1' (good) or '2' (probably good) are used and the measurements are cleaned by moving a 5x5 degree block to remove measurements more than 3 RMS from the mean.

To be able to average measurements monthly over a basin or a polygon, a grid is constructed by statistical interpolation of the steric sea levels at the profile locations based on the method described in Bretherton et al. (1976) and Gaillard et al. (2009). First, a background field is constructed by estimating a model through the 1000 closest measurements of a profile or grid cell location. This model contains a constant, a second-order 2-D lon-lat polynomial and six intra-annual to annual cycles (Roemmich and Gilson, 2009). The background sea surface height is taken as the model evaluated at the grid cell (or profile) location.

Consecutively, the background field vector $\hat{h}_{ssl,b}$ is subtracted from the sea level estimates, which results in:

$$\delta\hat{h}_{ssl} = \hat{h}_{ssl} - \hat{h}_{ssl,b},\tag{15}$$

where $\delta\hat{ssl}$ are the residuals. The ocean variance $\sigma_t^2$ is assumed to be 100 cm$^2$ (typical mesoscale variability (Chelton et al., 2007)), which is close to the average squared RMS-of-fit of the differences of the measurements with the model. These variances are subdivided into three components to represent different correlation scales as follows (Roemmich and Gilson, 2009):

$$\sigma_1^2 = 0.77\sigma_t^2$$
$$\sigma_2^2 = 0.23\sigma_t^2$$
$$\sigma_3^2 = 0.15\sigma_t^2\ ,\tag{16}$$

which are then used to construct covariance matrices $C(d)$ based on those used for the Scripps fields (Roemmich and Gilson, 2009), such that:

$$C(d) = \sigma_1^2 e^{-(\frac{d}{140})^2} + \sigma_2^2 e^{-\frac{d}{1111}},\tag{17}$$

and the measurement and representativity error matrix $R$:

$$R = diag(\sigma_3^2),\tag{18}$$





where $d$ is a measure for the distance between the profiles $p$ and the grid points $g$, such that:

$$d = \sqrt{a^2 d_x^2 + d_y^2}. \tag{19}$$

The parameter $a$ is 1 above 20 degrees latitude and below that it decays linearly to 0.25 at the Equator, in order to represent the zonal elongation of the correlation scale here (Roemmich and Gilson, 2009).

Using the correlations $C_{pg}$ (between profiles and grid points) and $C_p$ (between profiles), the weight matrix $K$ is computed as:

$$K = C_{pg}(C_p + R)^{-1}. \tag{20}$$

The weight matrix is then used to compute a vector of steric sea levels $\hat{s}_{ssl,g}$ for every grid point with in the area:

$$\hat{s}_{ssl,g} = K\delta\hat{h}_{ssl} + \hat{h}_{ssl,b}, \tag{21}$$

for which also the variance-covariance matrix $C_{ssla}$, so that:

$$C_{ssl,g} = C_g - KC_{pg}^T, \tag{22}$$

where $C_g$ are the correlations between grid locations.

In order to average the steric sea level anomalies, the values are weighted by the cosine of the latitude, which results in:

$$\omega_i = \cos(\lambda_i). \tag{23}$$

Like for altimetry, the weights are normalized:

$$w_i = \frac{\omega_i}{\sum_{i=0}^{I} \omega_i}. \tag{24}$$

Eventually these are used to compute the mean steric sea level $\bar{h}_{ssl,g}$ and its associated error $\bar{\sigma}_{ssl,g}$, with

$$\bar{h}_{ssl} = \hat{w}^T \hat{h}_{ssl,g}, \tag{25}$$

and

$$\bar{\sigma}_{ssl} = \sqrt{\hat{w}^T C_{ssl,g} \hat{w}}. \tag{26}$$

### 3.3 GRACE ocean bottom pressure

We use the full variance-covariance and normal matrices to filter the spherical harmonic coefficients with an Anisotropic Non-Symmetric (ANS) filter (Klees et al., 2008). This Wiener filter exploits the ratio between the variance of the error and of the signal to filter the coefficients. With the variance-covariance matrices $C_x$ and $D_x$, for respectively the errors and the signals,

the spherical harmonic coefficients $\hat{x}$ are filtered as:

$$\hat{x}_{of} = (C_x^{-1} + D_x^{-1})^{-1} C_x^{-1} \hat{x}. \tag{27}$$





For the filtered coefficients $\hat{x}_{of}$ a corresponding variance-covariance matrix $C_{x,of}$ is computed. This is a joint inversion of a static background field, which is set to zero, and the time-varying coefficients, resulting in:

$$C_{x,of} = (C_x^{-1} + D_x^{-1})^{-1}. \tag{28}$$

The derivation is elaborated in A.

The filtered grids contain ringing effects around strong signals over Greenland and the Amazon region, which can have substantial effects on the estimated trends in the ocean, which is discussed in Sect. 4. If averaged over large areas this will have hardly an affect, but on regional scales the ringing should be reduced. To obtain smoother fields, we use a fan filter (Zhang et al., 2009; Siemes et al., 2012), which is given as:

$$\hat{x}_{ff} = sinc(\frac{l}{l_{max}})sinc(\frac{m}{l_{max}})\hat{x}_{of}, \tag{29}$$

where the degree $l$ and order $m$ are related to the maximum degree, $l_{max}$. For a maximum degree of 60 and 96, this is comparable to a Gaussian filter of 280 km and 110 km, respectively (Siemes et al., 2012). Suppose $F_f = diag(sinc(\frac{l}{l_{max}})sinc(\frac{m}{l_{max}}))$, then the resulting covariance matrix $C_{x,ff}$ is written as:

$$C_{x,ff} = F_f C_{x,of} F_f. \tag{30}$$

Note that there is a fundamental difference between filtering the CSR and ITSG solutions. The CSR solutions are computed with respect to a static gravity field, while the ITSG solutions are computed with the respect to a static gravity field including a secular trend and an annual cycle. As a consequence, the CSR spherical harmonic coefficients and signal variance-covariance matrices include the annual and the secular trend while Wiener and fan filtering. The ITSG gravity fields are Wiener-filtered first, then the annual cycle and the secular trend are added back and eventually the fan filter is applied.

Since the degree-1 coefficients are not measured by GRACE, we add those of Swenson et al. (2008) to the CSR solutions. For the ITSG solutions a dedicated degree-1 solution is computed, using the same approach. Furthermore, we replace the $C_{20}$ coefficient with satellite laser ranging estimates (Cheng et al., 2013).

The intersatellite accelerations of GRACE are dealiased for high frequency ocean and atmosphere dynamics with the Atmospheric and Ocean De-aliasing Level-1B (AOD1B) product. Monthly averages of the AOD1B are provided as the GAD product

for both CSR and ITSG, where the mass changes over land are set to zero. To be able to combine GRACE OBP with inverse barometer corrected altimetry, the GAD products containing the modelled oceanic and atmosphere mass are added back in the form of spherical harmonics. Because the ocean model in the AOD1B product is made mass conserving by adding/removing a thin uniform layer of water to or from the ocean, the degree zero is removed before subtraction from the GAD product to compensate for the mean atmospheric mass change over the ocean, which is not measured by altimetry (Chambers and Willis,

2010).

To compute the OBP at a specific grid cell, the $4\pi$-normalized associated Legendre functions $\hat{y}$ are evaluated at its latitude-longitude location. The OBP of the grid cell $h_{obp}$ is consecutively calculated with:

$$h_{obp} = \hat{y}^T \hat{x}_{ff}. \tag{31}$$



For multiple grid cells the vector $\hat{y}$ becomes a matrix $Y$, such that $\hat{h}_{obp}$ becomes a vector of EWHs. This is written in form:

$$\hat{h}_{obp} = Y^T \hat{x}_{ff}. \tag{32}$$

It is possible to compute the grid's variance-covariance matrix $C_{obp}$ as (Swenson and Wahr, 2002):

$$C_{obp} = Y^T C_{x,ff} Y. \tag{33}$$

The averaging over an area is equal to that of the Argo grids. Suppose that $\hat{w}$ are the normalized latitude weights for the grid OBPs, then

$$\bar{h}_{obp} = \hat{w}^T \hat{h}_{obp} \tag{34}$$

is the mean ocean bottom pressure in and

$$\bar{\sigma}_{obp} = \sqrt{\hat{w}^T C_{obp} \hat{w}} \tag{35}$$

is its error.

Ultimately, the mean GIA OBP over a polygon (degree 2 and higher, as measured by GRACE) are subtracted from the results, to make them compatible with altimetry.

## 4   Comparison with existing products

In this section a comparison is made between existing products and the sea levels from altimetry, gravimetry and Argo floats.
First, we compare the MSL time series over the North Atlantic with the existing time series provided by the NOAA Laboratory for Satellite Altimetry (Leuliette and Scharroo, 2010) and we show the effect of a latitude dependent intermission bias. Secondly, amplitude and trend grids of steric sea level are compared to those computed from Scripps salinity and temperature grids (Roemmich and Gilson, 2009) and the Glorys reanalyses grids (Ferry et al., 2010). Thirdly, the optimally and fan filtered gravimetry grids are compared to the DDK5-filtered gravity fields (Kusche, 2007)(Kusche et al., 2009).

### 4.1   Total sea level

Fig. 4 shows a comparison of the NOAA time series with the ones computed in this study, for the North Atlantic above 30 degrees latitude. The NOAA time series were computed by averaging over 3x1 degree grid cells and then weighting them according to their latitude. The green, red and blue time series are all computed using the same geophysical corrections as given in the second column of Table 2, while to the light blue line the geophysical correction in the first column are applied.
As visible from the figure, hardly any differences are observed between all four time series. The RMS differences between all time series computed in this study and NOAA are in the order of 3-4 mm, which is slightly larger than differences found between the GMSL time series (Masters et al., 2012). The fact that the red and blue line resemble each other indicates that the underweighting of high-latitude measurements in the Nerem method hardly has any effect on the trend. This also holds



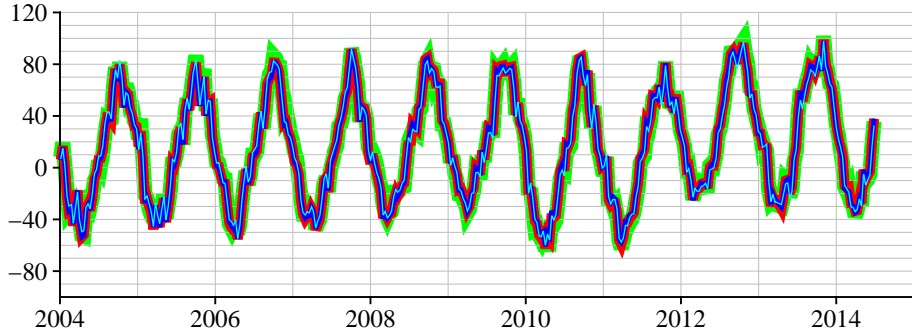

**Figure 4.** Comparison between North Atlantic mean sea level time series in mm of NOAA (green), the Nerem method (red), our method (blue) and our method using a geographically dependent intermission bias correction and the latest geophysical corrections (light blue).

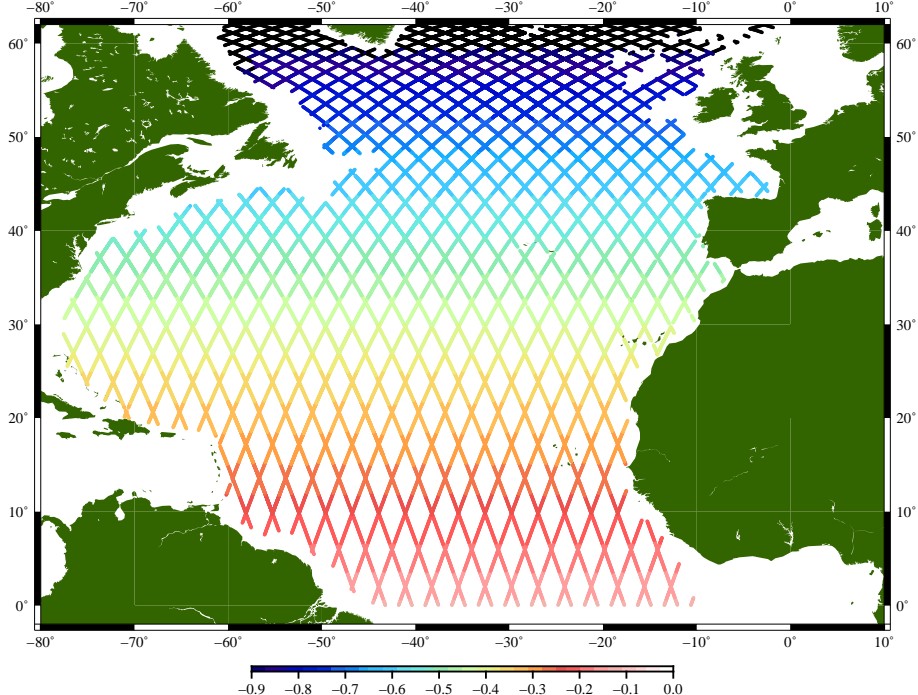

**Figure 5.** Differences in sea level trends in mm/y, computed with and without a latitude dependent intermission bias.

for averaging over smaller areas in the North Atlantic, where the only noticeable difference occurs when a substantial number of satellite tracks are missing, due to some maintainance or orbit manoeuvres. However, the time series may differ in places where strong decorrelations or strong differences in trends occur, especially in the North-South direction at high latitudes.

   The application of a latitude-dependent intermission bias has a substantial effect on the trend. From the NOAA time series a trend of 1.5 mm/year is found, while the time series from the Nerem and our method provide a trend of 1.8 mm/year. This difference can be explained by the combination of another averaging technique and geographically varying trends in sea



level. However, if the difference in MSL is computed between Jason-1 and Jason-2 over the North Atlantic during the tandem phase and this used as the intermission bias correction, trends of respectively 1.3 and 1.4 mm/year for the theoretical and the empirical weighting method are found. This is comparable to the trend computed by applying the geographical dependence of the intermission bias (the light blue line), which is 1.4 mm/year. To further illustrate this, Fig. 5 shows the differences in trend

if a constant intermission bias is used or a latitude dependent one. The mean difference of 0.4 mm/year is already significant, but locally the differences in trend increase to approximately 0.8 mm/year.

### 4.2 Steric sea level

In Fig. 6 grids for the amplitudes and trends of the steric signal are shown. The Scripps grids (Roemmich and Gilson, 2009) and our solution are solely based on Argo data, while the Mercator reanalyses product Glorys 2V3 assimilates various types

of data including altimetry (Ferry et al., 2010), sea surface temperature and Argo. Besides the different input data, the Glorys relies on a volume conserving ocean model, while the other two methods are statistical, the results can differ quite substantially. Since we use the same correlation structures as Scripps, the resulting grids should resemble each other quite closely. However, to be able to create a variance-covariance matrix between grid cells, it was required to do a 2D-interpolation of the steric sea levels instead of a 3D-interpolation of temperature and salinity profiles. In order to do this, the criteria for removing profiles

is, as described in Sect. 3.2 is different for both methods. As a consequence of the 2D-interpolation and the differences in the removal criteria the results differ.

In terms of the amplitudes of the annual signal, all three methods provide similar results in terms of the large scale features. Typically, large signals are found in the Gulf Stream region and close to the Amazon basin, while the areas around Greenland and West of Africa have small amplitudes. The Glorys grid differs from the others primarily in the Labrador sea and Northwest

of Ireland. Secondly, compared to Scripps the grids, the other two methods are slightly noisier. As long as the regions over which budgets are made are large enough, the methods will not differ substantially in terms of annual amplitude.

The plots in the right column of Fig. 6 reveal immediately a significant difference between the statistical methods and the reanalysis in terms of trend. It is not completely clear what the cause for this difference is. Since the Scripps grid and our grid resemble in terms of large scale features and are purely based on T/S-data, we trust the interpolation of Argo. The difference

between those two methods are again primarily the noise in the grids and the area around the Antilles, where Argo samples poorly as discussed in Sect. 2.

### 4.3 Ocean bottom pressure

In Fig. 7 the trends and amplitudes of the CSR DDK5-filtered gravity fields compared with those obtained from the Wiener-filtered ones from CSR and ITSG. Note that the Wiener-filtered solutions are also fan-filtered, as discussed in Sect. 3.3, but

will be referred to as Wiener-filtered from here on. Both in the annual amplitude and the trend grids some residual striping effects are present for the DDK5 solutions, yielding non-physical trend patterns in OBP. The Wiener filter strongly reduces the striping and as a result especially the trend grids are smoother. However, the ITSG grids also exhibit striping (as it appears at shorter wavelengths), which is the results of adding back the trend and annual cycle from the static field, as discussed in





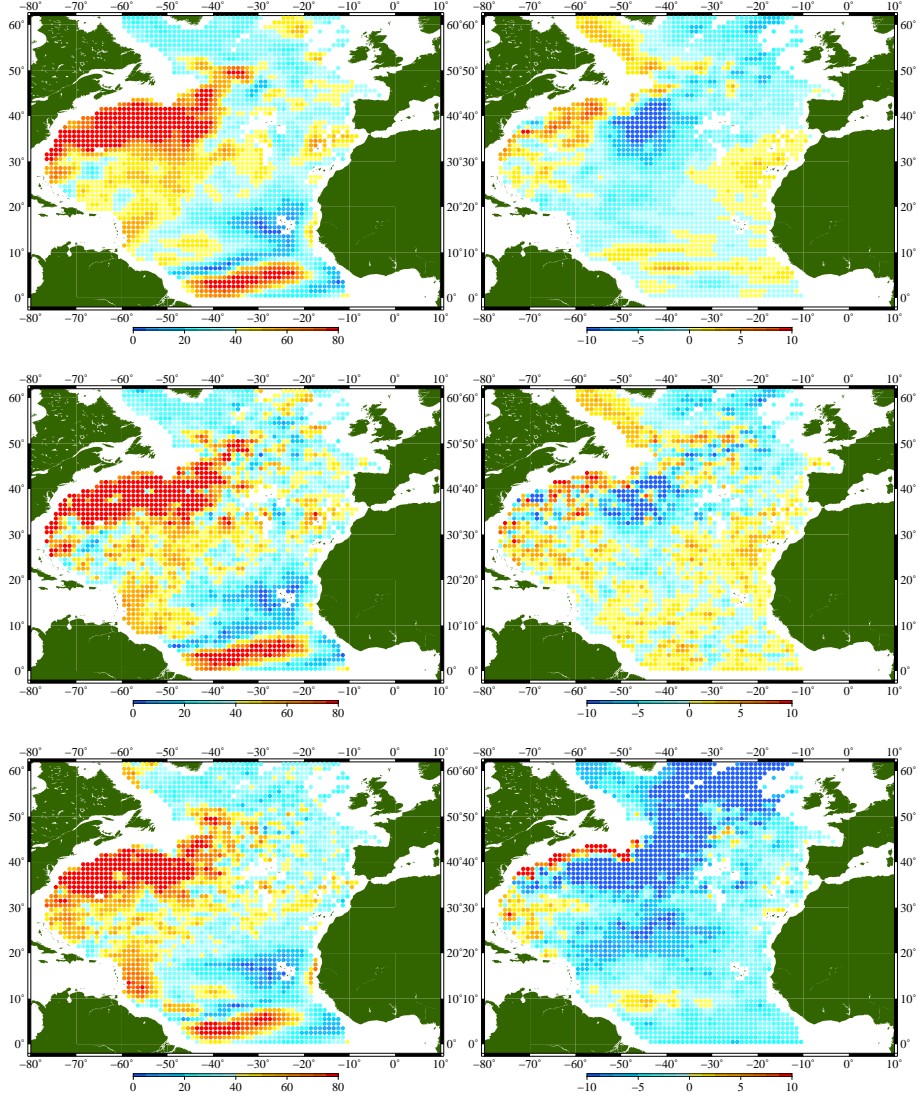

**Figure 6.** Amplitudes of the annual signal (left) in mm and trends (right) in mm/y computed with the Scripps grid (top), the method in this study (middle) and from the Glorys reanalysis product (bottom).

Sect. 3.3. A second observation is that the DDK5- and Wiener-filtered ITSG amplitudes are slightly larger, which indicates that a part of the annual signal is lost in the CSR Wiener-filtered solutions. Thirdly, Tamisiea et al. (2010) estimated a slight increase in OBP amplitude using fingerprint methods based on forward models of water mass redistribution around the Sahel and Amazon of 10-15 mm. In the Wiener-filtered CSR grids, also larger amplitudes are visible in these regions, however their

5  amplitude of 30-60 mm is far too large and are probably the result of hydrological leakage. This leakage is slightly reduced in the 96-degree solution compared to those of the 60-degree solutions.





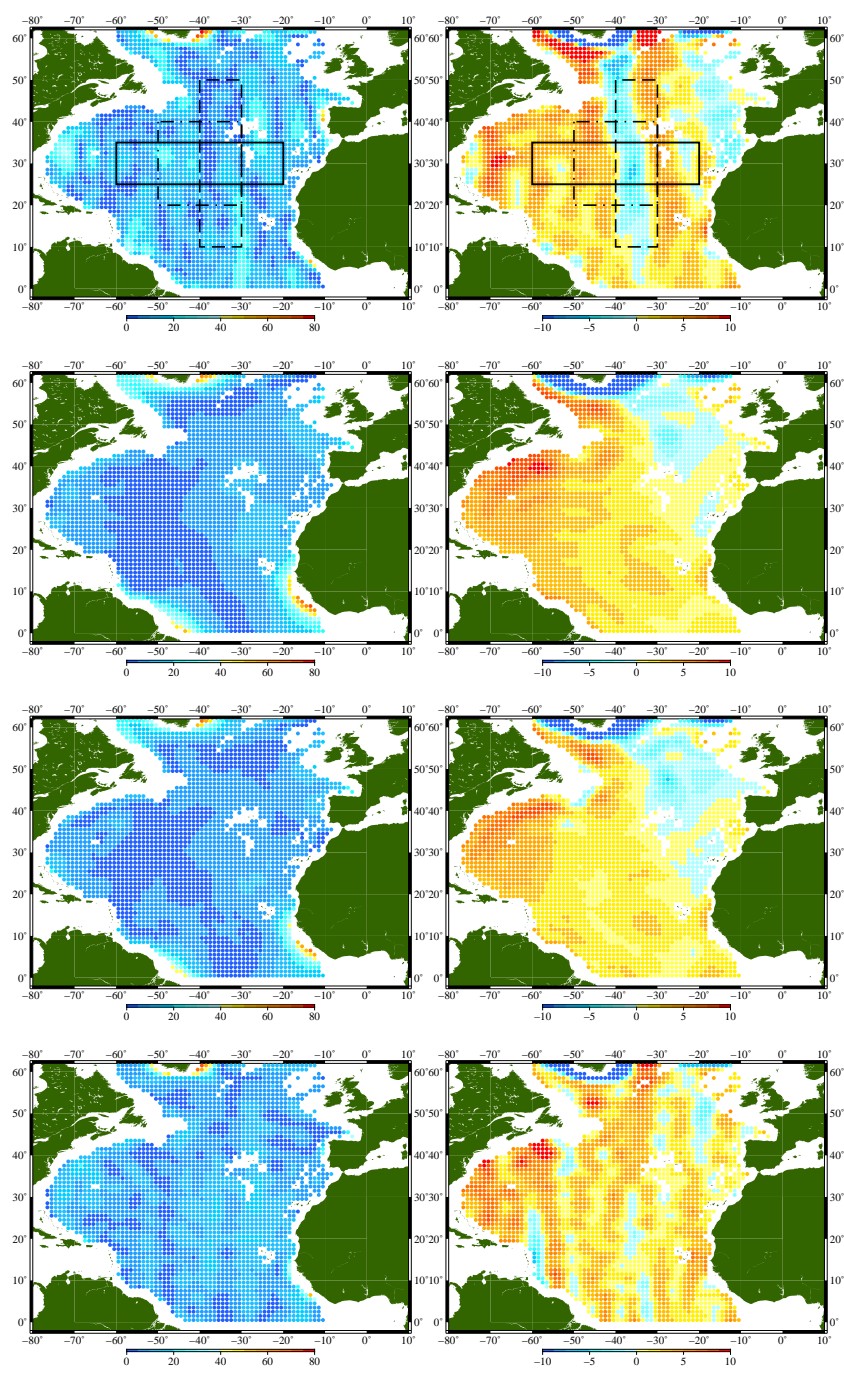

**Figure 7.** Amplitudes of the annual signal (left) in mm and trends (right) in mm/y of the OBP signal. The first to the third row show respectively the CSR DDK5-filtered solutions, the CSR 60- and 96-degree Wiener-filtered solutions. The fourth row shows the ITSG 90 degree Wiener-filtered solution.





To determine how this effects sub-basin scale OBP time series, it is first required to determine the minimum area over which the measurements have to be integrated. GRACE gravity fields have a resolution of typically 250-300 km half-wavelength (Siemes et al., 2012). For small ocean signals after applying filtering procedures, we expect the resolution to be closer to 400-500 km. Argo has approximately one to two floats per $3 \times 3$ degree box, so its resolution is in the same range as that of GRACE. Jason-1&2 have an inter-track spacing of 315 km at the Equator, which decreases substantially at 60 degrees latitude. Considering all systems, this theoretically makes it possible to create budgets over grid cells approximately 500x500 km, however due to the limited length of the time series, the error bars on the trends become much larger than the signals. The size of the polygons is therefore chosen based on the criterion that the error on the trends does not exceed 1 mm yr$^{-1}$.

To illustrate the effects of different filters and residual striping on sub-basin scale budgets, Fig. 8 shows time series of OBP averaged over the polygons shown in Fig. 7. All three polygons have approximately the same size, but have different orientations. Except for the months surrounding the near 4-day repeat-period in 2004, where the variance-covariance matrices of CSR probably do not properly described the noise of the gravity fields, the time series resemble each other best for the zonally oriented polygon. In the zonal polygon the noise in the CSR 96-degree Wiener-filtered solution is substantially larger than the other results. Futhermore, it becomes clear that the DDK5-filtered solutions do not contain substantial signal above degree 60, because the red and yellow lines are on top of each other, while the Wiener-filtered solutions are substantially different.

The month-to-month noise of the CSR Wiener-filtered time series is comparable for all three polygons. The DDK5-filtered time series becomes much noisier for the meridionally oriented polygon, where month-to-month jumps of 10-20 mm occur. In addition, the DDK5 time series exhibit a substantially different trend in the meridional polygon than the other time series, because the orientation of the polygon is aligned with the residual stripes (Fig. 7). So in terms of trend and noise the DDK5 time series strongly depends on the orientation of the polygon. Eventhough the ITSG trend and amplitude grids suffer from striping, they do not become significantly noisier for the meridionally oriented polygon.

## 5 Results and discussion

The first objective of this section is to reveal patterns of sea level amplitudes and trends in the North Atlantic and how these resemble for the two different methods: altimetry and Argo+GRACE. Secondly, this section discusses the closure of sea level budgets over polygons of approximately one-tenth of the North Atlantic in terms of trend, annual amplitude and residual variability. It is shown for which regions the budget is closed and possible causes for non-closure are discussed. Thirdly, we focus on the best choice of GRACE filter solutions for the OBP component.

### 5.1 North Atlantic sea level patterns

In Fig. 9 grids of trends and amplitudes computed from Argo+GRACE are overlaid with Jason derived trends and amplitudes at the ground-tracks. In areas where the ground-tracks of altimetry are barely visible, there is a good resemblance between Argo+GRACE and altimetry.





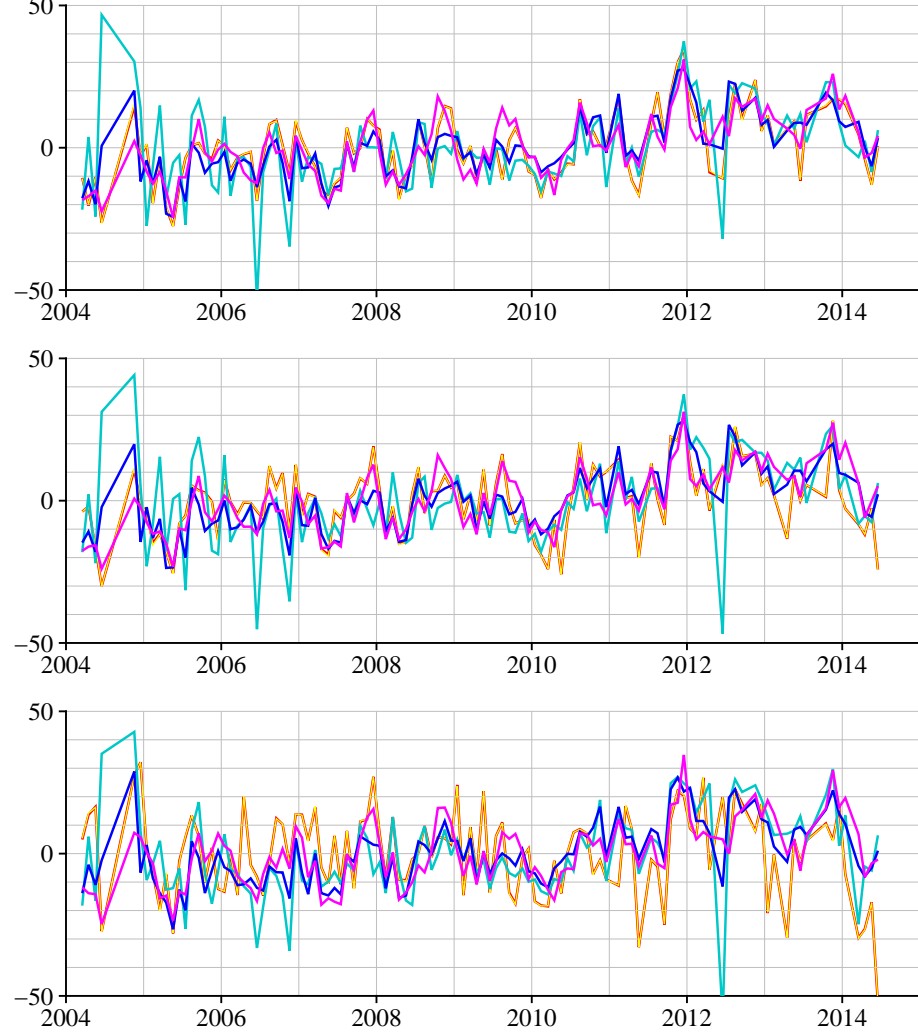

**Figure 8.** Sub-basin scale time series of OBP in mm using various filters for three polygons with different orientation: landscape (top), square (middle) and portrait (bottom). Red and yellow represent the CSR DDK5-filtered solutions cut-off at 60 degree and 96-degrees. The blue and light-blue time series represent respectively the Wiener-filtered CSR 60- and 96-degree solutions. In purple are the time series of the Wiener-filtered ITSG solution.

The grids and ground-tracks shown in the left column show that large annual signals are present in the Gulf Stream region and in a tongue stretching from the Amazon to the Sahel. A region without any substantial annual signal is located just West of Africa, which is clearly visible in both the Argo+GRACE grid and altimetry. Both methods reveal these large-scale oceanographic features in amplitude, but there are also quite some differences. East of the Antilles, altimetric measurement show an annual amplitude of more than 60 mm, whereas Argo+GRACE estimates are in the range of 40-50 mm, depending on the choice of GRACE filter. Note that in this area, there are barely any Argo floats (Fig. 1), which might lead to interpolation



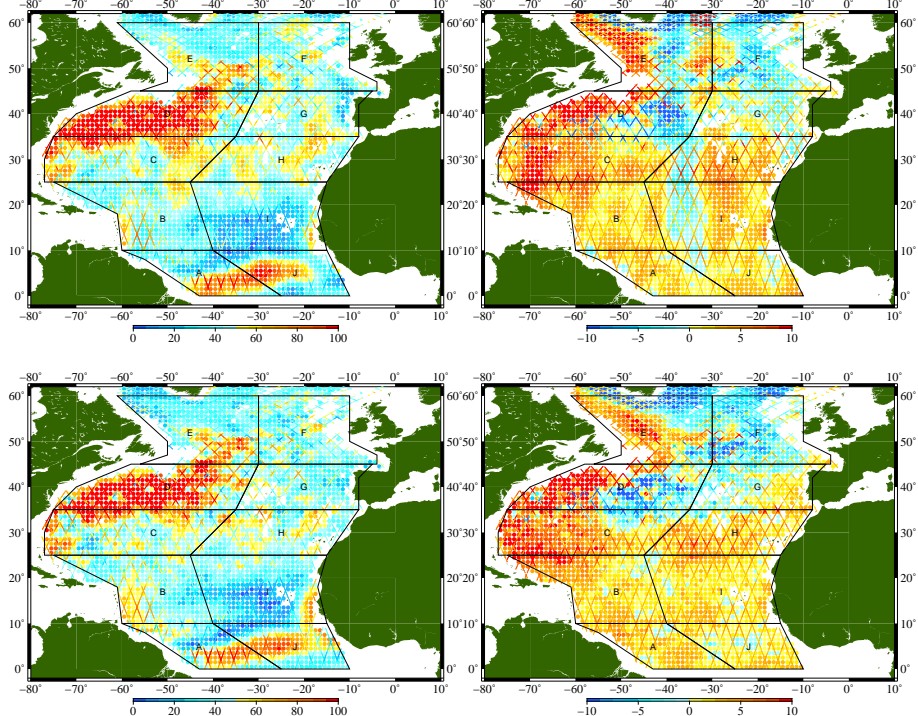

**Figure 9.** Amplitudes of the annual signal (left) in mm and trends (right) in mm/y computed of the sum of the components (Argo+GRACE) overlayed with those computed from the total sea level measured with altimetry. For the two top figures the CSR DDK5-filtered solutions are used and for the bottom the Wiener-filtered CSR solution, both up to d/o 96.

problems. A second difference is observed in the Wiener-filtered grid (bottom-left) at the Amazon and Sahel regions. This is exactly at the areas where the Wiener-filtered OBP grids of Fig. 7 exhibit probable hydrological leakage.

The trends from altimetry in the right column of Fig. 9 show a distinct pattern, where positive trends are found below 35 degree latitude and negative trends above this line, with the exception of the North-American coastline. Large trends along the
5  North-American coast are also found by tide gauge studies (Sallenger at al., 2012), where they attribute this to a weakening Atlantic Meridional Overturning Circulation (AMOC). The Wiener-filtered Argo+GRACE solution resemble the trend patterns derived from altimetry measurement better, while the residual stripes in the DDK5 solution are clearly visible. Note that a significantly larger altimetric trend is visible in polygon H, just in front of the Mediterranean. Possible causes will be discussed later.

10  **5.2    Sub-basin scale budgets**

The North Atlantic is split into ten regions, divided in the middle by the Mid-Atlantic ridge, while in the latitude direction trying not to cut through the major oceanographic features, like the salt water tong in front of the Mediterranean and the Gulf





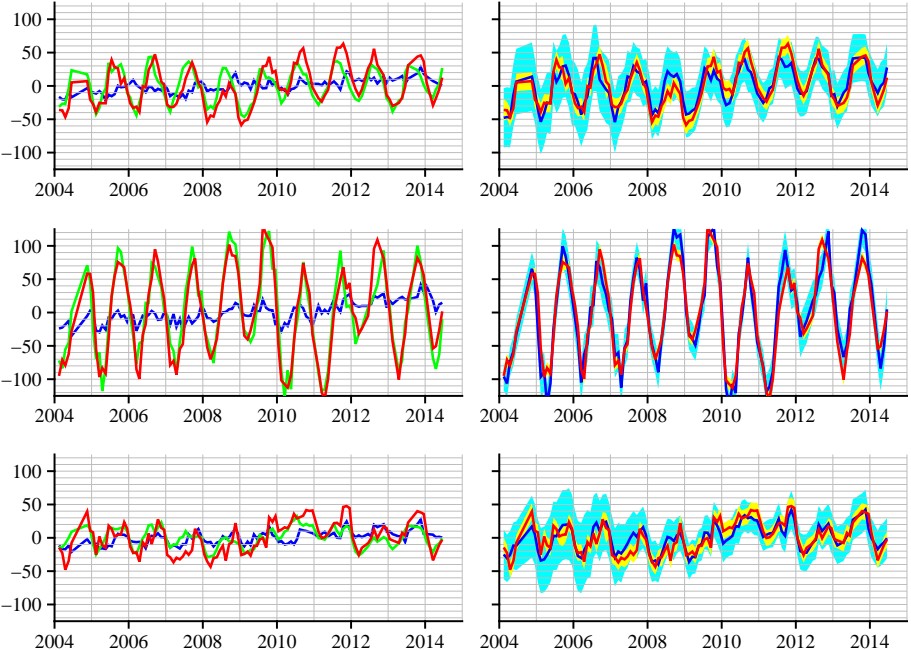

**Figure 10.** Time series of sea level (mm) components for regions B (top), D (middle) and I (bottom) Left: total sea level from altimetry in red, steric sea level in green and the ITSG OBP in blue. Right: total sea level from altimetry in red and the sum of the two components in blue. In light blue and yellow their 95 % confidence interval.

Stream, as shown in Fig. 9. As in Sect. 4.3, the size of basins is chosen based on the criterion that the error on the does not exceed 1 mm yr$^{-1}$. First, we will discuss three representative examples of time series. Then budget closure in terms of trend and annual amplitude is addressed and the corresponding best gravity filter is determined. Ultimately, the trends, semi-annual and annual signals are reduced from the time series and the best filter choice in terms of residual variability is determined.

*Timeseries*

Budgets for three representative regions, using the Wiener-filtered OBP solutions, are shown in Fig. 10. The time series for the rest of the regions can be found in the supplementary material. The left plots confirm that the main driver for annual fluctuations in sea level is the steric sea level, but that the trend is strongly influenced by a mass component. On the right side we see that the

10 sum of the components and the total sea level agree to within the error bars, but some problems arise in the Gulf Stream area (region D), probably caused by sharp gradients in sea level. The sea levels in Polygons D and I also contain some interannual signals, which is especially pronounced between 2010-2012. The left column shows that the interannual variability is primarily a steric signal. Note that the larger size of the error bars in regions B and I is due to the decrease in altimetry track density closer to the Equator and the elongation of the correlation radius for the interpolation of Argo floats.



*Trends*

Trends computed from the time series of Fig. 10 are given in Table 2. Close to the Equator (A, B, I and J) and over the whole North Atlantic the trend budget is closed within two standard deviations no matter which of the OBP solutions is used. This confirms the results of Fig. 9. In this region the GIA correction is relatively small and no sharp gradients or strong features

are present in the trend grids, which contribute to proper budget closure. Budget closure is also achieved by all filters in the northeast of the Atlantic (F, G). Again, this is a relatively quiet region, with no significant gradients in trends and a small GIA correction. The results of the Argo+GRACE however show are larger spread for the different OBP solutions. It is important to note that especially region F suffers from some ringing artefacts before the fan filter is applied and that the far northeast is not very well covered by Argo floats. The trends of the DDK5-filtered solution in region G are a bit further off than the other

solutions, probably resulting from the striping as visible in Fig. 7.

In the northwest of the Atlantic the choice of gravity field filter either substantially influences the estimated trends (D and E), or they are just outside of two standard deviations (C) for one ore more solutions. Using the 96-degree CSR Wiener filtered solution, the budget is closed within two standard deviations for all three polygons, but the others do not. For region C, the results of the all filters are resembling quite well, but some are just outside of two standard deviations from altimetry. For region

D, the 60-degree CSR Wiener-filtered results are far off, but the other results are close again. In this region, sharp gradients occur not only in the OBP component with the presence of a neighbouring continental shelf, but also in the steric component. This might lead to leakage of the continental shelf OBP signal or problematic interpolation of the Argo steric sea levels. In addition for both of the beforementioned regions, the GIA correction on the OBP is relatively large. Adding a GIA correction error of 10-20 %, which is smaller than discussed in Sect. 3.3, to the OBP trends would close the budget in these regions for all

the solutions, except for the CSR 60-degree solution in region D. In region E, a clear split is visible between the Wiener-filtered CSR solutions, which close the budget and the other two solutions, which do not close the budget. The difference in results could be caused by the filter not being able to handle the large gradients (Klees et al., 2008) in OBP within this region (Fig. 9). However, if we would again add only a 10-20 % GIA correction error, it would suffice to close the budget for all filters.

Ultimately, only the budget in region H cannot be closed with any of the solutions and there is no strong GIA signal present,

which could be responsible for a large bias. In addition, the sea level in this polygon does not exhibit any strong gradients and the number of Argo floats is substantial. This excludes interpolation or filtering problems. Therefore, we argue that this can be explained by a deep-steric effect, that could be related to variations in the export of saline water from the Mediterranean (Ivanovic et al., 2014), which is not captured by Argo.

In conclusion, it is possible to close the sea level budget within two standard deviations for nine-out-of-ten regions using

the Wiener-filtered 96-degree solutions. If a 10-20 % GIA correction error is taken into account, the budget for nine-out-of-ten polygons is also closed for the DDK5-filtered solution and the Wiener-filtered ITSG solution. This also suggests that the commonly assumed GIA correction error of 20-30 % (Von Schuckmann et al., 2014) is probably overestimated.



**Table 2.** Trends of total sea level (mm yr$^{-1}$) and their standard deviations from altimetry (Jason) and the sum of steric and OBP from Argo (A.) and GRACE (CSR, ITSG) for different filter solutions. NA is the trend for the complete North Atlantic between 0-65 degree latitude. A 0.4 mm y$^{-}$1 drift error is taken into account for altimetry based on the comparisons with tide gauges (Mitchum, 1998, 2000). *GIA Absolute Sea Level (ASL) correction subtracted from altimetry MSL and GIA EWL correction subtracted from GRACE OBP.

|     | Jason       | CSR96+A. DDK5 | CSR96+A. Wiener | CSR60+A. Wiener | ITSG90+A. Wiener | GIA ASL | GIA EWL |
| --- | ----------- | ------------- | --------------- | --------------- | ---------------- | ------- | ------- |
| A   | 2.6±0.5     | 1.8           | 2.4±0.9         | 2.7±0.9         | 2.3±0.9          | -0.3    | -2.2    |
| B   | 2.8±0.5     | 3.1           | 3.0±0.7         | 3.7±0.7         | 3.1±0.7          | -0.5    | -3.4    |
| C   | 3.2±0.4     | 4.2           | 4.4±0.5         | 4.8±0.5         | 4.5±0.5          | -0.6    | -5.1    |
| D   | 1.0±0.4     | 1.5           | 1.9±0.5         | 3.1±0.5         | 2.3±0.4          | -0.6    | -6.0    |
| E   | 0.5±0.4     | 2.2           | 0.3±0.5         | 0.0±0.5         | 2.2±0.4          | -0.5    | -7.1    |
| F   | -2.4±0.4    | -2.0          | -3.4±0.5        | -3.0±0.5        | -1.8±0.4         | -0.5    | -4.6    |
| G   | 0.7±0.5     | -0.8          | -0.2±0.6        | 0.0±0.6         | 0.4±0.6          | -0.5    | -3.6    |
| H   | 4.7±0.4     | 1.4           | 2.5±0.6         | 2.7±0.6         | 3.3±0.6          | -0.5    | -3.5    |
| I   | 2.3±0.4     | 1.4           | 2.1±0.6         | 2.5±0.6         | 2.5±0.6          | -0.5    | -2.8    |
| J   | 2.4±0.4     | 1.7           | 1.3±0.7         | 1.3±0.7         | 1.6±0.6          | -0.3    | -2.0    |
| NA  | 1.8±0.4     | 1.8           | 1.5±0.3         | 1.8±0.3         | 2.2±0.2          | -0.5    | -4.1    |

*Annual signal*

We indicated that the seasonal cycles are primarily caused by steric variations in sea level (Fig. 10). By comparing the first column with the last column in Table 3, it becomes clear that in most cases an additional OBP signal is required to close the budget in terms of annual amplitude. The discrepancy between Argo and altimetry for the whole North Atlantic reveals that on average in-phase OBP signals with an amplitude of approximately 7 mm are required to close the budgets, which is in line with the modelled results of Tamisiea et al. (2010). They modelled, using fingerprints, amplitudes of OBP ranging from 3-12 mm, and phases (not shown here) between day 210-330, which is in-phase with the steric signal.

Table 3 shows that for virtually every region the choice of filter matters. On top of this, there is a clear difference between the Wiener-filtered CSR solutions and the other two solutions. Adding the Wiener-filtered CSR solutions increases in a few cases even the discrepancy with altimetry, which is caused by an out-of-phase OBP signal. Especially in regions A and J, where the amplitude is respectively underestimated and overestimated. Only in four regions (C, D, H and I) the amplitude budget closes within two standard deviations using these solutions.

Even though no error bars are computed for the DDK5-filtered solutions, it is clear that the results are far better in terms of budgets closure. The results are comparable the Wiener-filtered ITSG solutions, which closes seven-out-of-ten budgets within two standard deviations. CSR DDK5+Argo underestimates the amplitude in region B, while ITSG+Argo overestimates the amplitude with respect to altimetry in region D. In region B the estimate of ITSG+Argo is relatively small and in region D the CSR DDK5+Argo also relatively large. Note that the number of Argo floats in region B is often small (Fig. 1) and that





**Table 3.** Amplitudes (mm) of the annual signal from total sea level from altimetry and the sum of steric and OBP from Argo and GRACE for different filter solutions.

|     | Jason | CSR96+A. | CSR96+A. | CSR60+A. | ITSG90+A. | A. only |
|-----|-------|----------|----------|----------|-----------|---------|
|     |       | DDK5     | Wiener   | Wiener   | Wiener    |         |
| A   | 42.3±1.3 | 36.0 | 26.8±3.4 | 28.2±3.4 | 36.3±3.1 | 32.2±3.1 |
| B   | 34.2±0.9 | 27.5 | 27.8±2.7 | 29.6±2.7 | 30.5±2.5 | 30.2±2.4 |
| C   | 54.0±0.7 | 52.6 | 49.3±2.1 | 48.5±2.1 | 52.9±1.9 | 47.1±1.9 |
| D   | 82.1±0.6 | 85.0 | 84.3±2.0 | 82.8±1.9 | 88.3±1.7 | 82.6±1.7 |
| E   | 48.0±0.5 | 43.2 | 40.2±1.9 | 38.5±1.8 | 42.8±1.5 | 39.3±1.4 |
| F   | 45.8±0.6 | 40.4 | 37.6±2.0 | 39.6±1.9 | 41.2±1.6 | 35.1±1.6 |
| G   | 45.1±0.9 | 44.5 | 37.7±2.2 | 39.9±2.1 | 43.2±2.0 | 38.4±1.9 |
| H   | 49.9±0.8 | 48.8 | 45.1±2.3 | 46.5±2.3 | 48.1±2.1 | 39.6±2.1 |
| I   | 18.7±0.8 | 19.0 | 16.0±2.3 | 17.8±2.2 | 19.1±2.0 | 11.9±2.0 |
| J   | 40.3±1.2 | 40.8 | 46.1±2.5 | 49.0±2.4 | 42.9±2.2 | 33.9±2.1 |
| NA  | 44.6±0.3 | 42.6 | 39.5±1.1 | 40.0±1.0 | 43.3±0.8 | 37.7±0.8 |

large gradients in the steric sea level in region D could cause interpolation problems for steric sea level. Secondly, in both northern polygons E and F both combinations of Argo+GRACE underestimate the amplitude compared to altimetry. Why this underestimation occurs is not completely clear. A likely culprit is the gravity field filtering, but yearly deep convection events in these regions (Våge et al. , 2009), which transport surface water to depth below 1000 m, and the limited number of Argo

5 floats, could also be contributing factors.

Using the ITSG solutions it is also possible to close the budget on the scale of the whole North Atlantic (last row of Table 3). The Argo+ITSG performs best in terms of amplitude budget closure in most regions, eventhough often characterized by sligthly smaller amplitudes than those derived from altimetry. This suggests that there is either a long-wavelength underestimation of the amplitude in GRACE, an over estimation in altimetry, or a missing steric effect in Argo. This is in line with Storto et al.

10 (2015), where on a global scale, steric sea levels computed from reanalyses and gridded T/S fields are found to be smaller than those indirectly derived from altimetry-GRACE. Additionally, Marcos et al. (2011) found differences in phase and amplitude of steric-corrected altimetry and OBP from destriped 500 km Gaussian-filtered GRACE solutions in the North Atlantic.





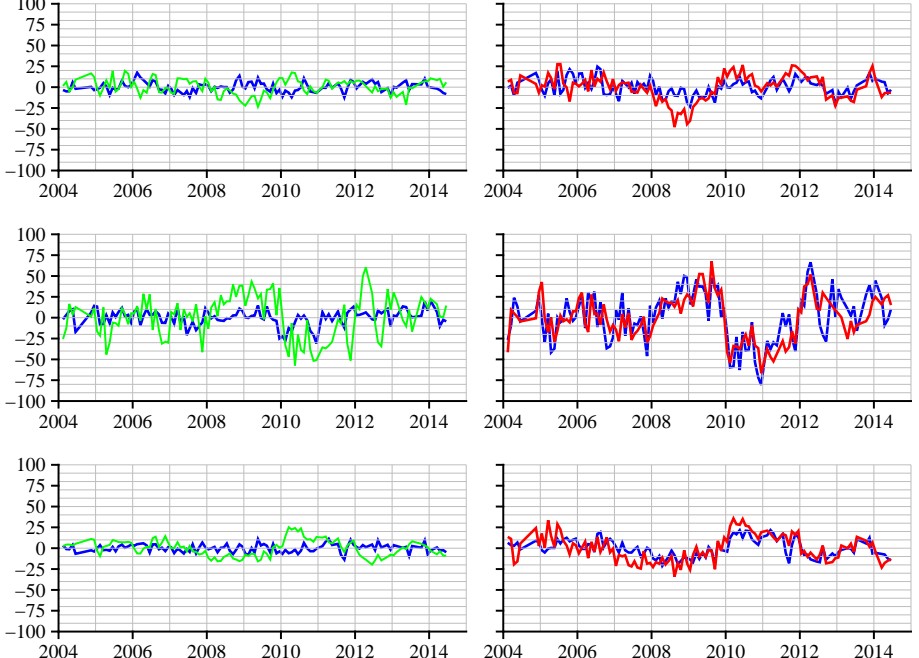

**Figure 11.** Time series of sea level (mm) components for polygons B (top), D (middle) and I (bottom) after removing the trends and the annual and semi-annual signals. Left: ITSG OBP in blue and steric sea level in green. Right: total sea level from altimetry in red and the sum of the two components in blue.

*Residual variability*

Time series for the same regions as in Fig. 10 are shown in Fig. 11, but their trend, semi-annual and annual signals have been reduced to show the residual variability. For the rest of the regions plots of the residuals are given in the supplementary material. In contrast to the time series for the whole Atlantic (not shown), the sub-basin scale time series show significant interannual

variability. Region D, located at the East Coast of the United States, shows a jump of 60-70 mm within three months at the end of 2009. This jump coincides with a shift in the Gulf Stream described by Pérez-Hernández and Joyce (2014) as the largest in the decade, which they related to the North Atlantic Oscillation. As illustrated in the left column, the shift in the Gulf Stream is primarily of steric nature, however small deviations in the OBP signal are also present. It is remarkable that at the same time on the other side of the Atlantic (region I, bottom figures), an increase in sea level is observed by both altimetry and Argo.

This suggest a link between the latitude of the Gulf Stream and sea level temperatures in the East of the Atlantic. In region B, we also observe a small interannual effect by altimetry and Argo. However, the amplitude of the signal is larger for altimetry than is captured by Argo, which suggests either some interpolation issues in an area without many Argo floats or a deep-steric effect.

Using any of the filtered CSR or ITSG solutions, it is possible to detect the interannual variability described, probably

because most of the signal is of steric origin. However, for the interannual signals that are less pronounced, or for high frequency



**Table 4.** Fraction of explained variance, $R^2$, of altimetry total sea level by Argo+GRACE steric+OBP for different gravity field filter solutions after removing the semi-annual and annual signals and the trend.

|     | CSR96+A. | CSR96+A. | CSR60+A. | ITSG90+A. |
|-----|----------|----------|----------|-----------|
|     | DDK5     | Wiener   | Wiener   | Wiener    |
| A   | 0.32     | 0.07     | 0.33     | 0.38      |
| B   | 0.02     | -0.46    | 0.09     | 0.24      |
| C   | 0.37     | 0.14     | 0.38     | 0.40      |
| D   | 0.31     | 0.16     | 0.36     | 0.34      |
| E   | 0.14     | -0.19    | 0.29     | 0.44      |
| F   | 0.09     | -0.17    | 0.45     | 0.52      |
| G   | 0.13     | -0.05    | 0.29     | 0.34      |
| H   | -0.12    | -0.49    | 0.21     | 0.27      |
| I   | 0.34     | 0.14     | 0.50     | 0.49      |
| J   | 0.39     | 0.17     | 0.45     | 0.53      |
| NA  | -0.05    | -1.21    | -0.06    | -0.01     |

behaviour of sea level there are some differences between the OBP solutions. Table 4 shows the fraction of variance of the residual signal of altimetry (trend, semi-annual and annual cycles removed) explained by Argo+GRACE.

The third column indicates that Argo in combination with Wiener-filtered 96-degree CSR solution does not explain much of the residual variance, but mostly introduces additional noise, which causes the negative values. Using the DDK5-filtered OBP

the explained variance increases, but the best performance is obtained with the Wiener-filted 60-degree CSR and especially the 90-degree ITSG gravity fields. The last column shows that after reducing the trend, and the semi-annual and annual signals, between 24-53 % of the residual signal can be explained by the combination of Argo and ITSG. It is remarkable that for the whole North Atlantic (last row), no variance is explained by the Argo+GRACE, primarily due to the absence of a clear interannual signal.

**6  Conclusions**

For the first time it is shown that sea level budgets can be closed on a sub-basin scale. With the current length of the time series it is possible to establish budgets over areas of approximately one-tenth of the North Atlantic. To obtain error bars on the annual amplitudes, trends and time series, errors for altimetry and Argo profiles are propagated from existing correlation functions, while for GRACE full variance-covariance matrices are used. For altimetry, a latitude dependent intermission bias is applied

and it is shown that this leads to trend differences ranging up to 0.8 mm yr$^{-1}$ if the period from 2004-2014 is considered.

To obtain proper averaged OBP for sub-basin scale polygons, the gravity fields have to be filtered. The application of an anistropic Wiener filter on 96-degree CSR solutions leads to the best closure of the budget in terms of sea level trends, with



closure in nine-out-of-ten regions. In the considered regions also the DDK5-filtered CSR solutions and the Wiener-filtered ITSG solution appear to close just as many budgets when a 10-20 % GIA correction error is added. The results of the DDK5 filter however, strongly depend on the orientation of averaging area due to residual meridional striping. The strong resemblence between trends also indicates that the errors on the GIA model are probably smaller than the commonly assumed 20-30 %.

Furthermore, a large difference in trend between altimetry and Argo+GRACE is observed in front of the Mediterranean where only a small GIA correction is applied. We believe that this originates from steric effects below the considered 1000 m, where saline water enters the Atlantic from the Strait of Gibraltar and dives to large depths. Further research is needed to confirm this hypothesis.

  The 60- and 96-degree Wiener-filtered CSR solutions appear to underestimate the amplitude of the annual signal substan-

tially. They also suffers from what appears to be leakage around the Amazon and Sahel, regions with a substantial annual hydrological cycle. Using the DDK5-filtered gravity fields and the ITSG solutions, the sum of the steric and OBP components becomes significantly closer to that of altimetry, with closure in seven-out-of-ten regions. However, it must be noted that the altimetry signals tend to be slightly larger. This is likely due to partly destruction of the signal by filtering of the gravity fields or limited Argo coverage, or in some regions deep-steric signals.

By removing the semi-annual and annual signals and trends interannual variability can be detected. Since most of the interannual variability in the North Atlantic is contained in the steric component, the type of filter on the gravity fields is not really important. However, if we look at differences on a month-to-month basis, high-frequency variations or small interannual fluctuations in OBP, it is best to use the 60-degree anisotropic Wiener-filtered solutions or the ITSG solutions, because the fraction of explained variance of the altimetric sea level time series by the sum the components using these solutions is largest.

Using the ITSG solution, 24-53 % of the variability in the altimetry-derived sea level time series is explained. The 96-degree Wiener-filtered CSR solution only introduces noise and explains virtually no residual variability of the altimetry time series. Especially in the 4-day repeat-orbits in 2004 and even the months around them, the Wiener-filtered solutions do not give proper estimates of the OBP components, which partly contributes to a lower explained variance.

  To summarize, using the ITSG Wiener-filtered solution the trend budgets close when a GIA correction of 10-20 % is applied.

They perform, together with the standard DDK5-filtered CSR solution, best in terms of annual amplitude budget closure. Additionally, the combination of ITSG OBP and Argo steric sea levels explains the largest fraction of variance in altimetry time series. Based on this, the best option to establish budgets, at scales considered in this paper, is the Wiener-filtered ITSG solution. However, due to residual striping in the trend grids from the 'static' background field that are added back after Wiener-filtering, one must take care when averaging OBP over even smaller regions, or meridionally oriented polygons, which is a

even a bigger problem for the standard DDK5-filtered CSR solutions.

## Appendix A

The Wiener filter is in principle a joint inversion between the spherical harmonic coefficients of the background field $\hat{x}_b$ and those of the time-varying gravity field $\hat{x}$. Suppose that $C_x$ is the error variance-covariance matrix of $\hat{x}$ and $D_x$ the signal



variance-covariance matrix, then the filtered coefficients $\hat{x}_f$ are expressed as:

$$\hat{x}_f = (C_x^{-1} + D_x^{-1})^{-1} C_x^{-1} \hat{x} + (C_x^{-1} + D_x^{-1})^{-1} D_x^{-1} \hat{x}_b. \tag{A1}$$

Assuming the spherical harmonic coefficients of the background field are zero, this equation reduces to Eq. 27. Its filtered variance-covariance matrix $C_{x,f}$ is computed using:

$$
\begin{aligned}
C_{x,f} = {}& (C_x^{-1} + D_x^{-1})^{-1} C_x^{-1} C_x ((C_x^{-1} + D_x^{-1})^{-1} C_x^{-1})^T \\
& + (C_x^{-1} + D_x^{-1})^{-1} D_x^{-1} D_x ((C_x^{-1} + D_x^{-1})^{-1} D_x^{-1})^T .
\end{aligned}
\tag{A2}
$$

Since the matrices $(C_x^{-1} + D_x^{-1})^{-1}$ and $C_x^{-1}$ are symmetric, it is possible to simply change the order underneath the transpose sign and leave:

$$
\begin{aligned}
C_{x,f} = {}& (C_x^{-1} + D_x^{-1})^{-1} C_x^{-1} C_x C_x^{-1} (C_x^{-1} + D_x^{-1})^{-1} \\
& + (C_x^{-1} + D_x^{-1})^{-1} D_x^{-1} D_x D_x^{-1} (C_x^{-1} + D_x^{-1})^{-1} ,
\end{aligned}
\tag{A3}
$$

which is further simplified by using the identity $C_x^{-1} C_x = I$ to:

$$
\begin{aligned}
C_{x,f} = {}& (C_x^{-1} + D_x^{-1})^{-1} C_x^{-1} (C_x^{-1} + D_x^{-1})^{-1} \\
& + (C_x^{-1} + D_x^{-1})^{-1} D_x^{-1} (C_x^{-1} + D_x^{-1})^{-1} .
\end{aligned}
\tag{A4}
$$

Finally, this equation is rewritten, such that:

$$C_{x,f} = (C_x^{-1} + D_x^{-1})^{-1} (C_x^{-1} + D_x^{-1}) (C_x^{-1} + D_x^{-1})^{-1}, \tag{A5}$$

which eventually is simplified to Eq. 28.

*Acknowledgements.* We would like to thank ITSG and CSR for providing their gravity fields including full normal and/or variance-covariance matrices. The Argo profile data are kindly provided on the website of the US National Oceanographic Data Center (NODC). We appreciate the service of the employees of Mercator for delivering the Glorys reanalysis product. We wish to express our gratitude to the RADS teams for constantly maintaining and updating their database. This study is funded by the Netherlands Organisation for Scientific Research (NWO) through VIDI grant 864.12.012 (Multi-Scale Sea Level (MuSSeL)).





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
