# Peer review of "Sub-basin scale sea level budgets from satellite altimetry, Argo floats and satellite gravimetry: a case study in the North Atlantic Ocean"

_Ocean Science, 2016_

## Referee Comment (RC1) · Anonymous Referee #1 · 20 Aug 2016

Review of Sub-basin scale sea level budgets from satellite altimetry, Argo floats and satellite gravimetry in the North Atlantic by Marcel Kleinherenbrink, Riccardo Riva, and Yu Sun. Submitted to Ocean Science.

Reviewed on 19 August 2016.

General comments The paper attempts merging sealevel from altimetry with GRACE gravity fields and steric height from Argo floats to close the sea level budget of sub-basin scale areas in the North Atlantic. Different GRACE products are intercompared with convincing results, both in seasonal and interannual variability and trends. The analysis is thorough and worth publication. In some respect the paper appears quite technical, and it should be noted that main aim of the study is to determine which

GRACE product and data processing method is most appriate to achieve the desired result, i.e. agreement of observed sealevel (from altimetry), and calculated sealevel from mass changes (GRACE) and steric changes (from Argo floats). The selection of the study area (North Atlantic) and its subdivision into smaller polygons is somewhat arbitrary. I was not able to proofread all equations stated by the authors, I assume that all calculations have been made correctly. I have a few general comments, and a number of minor comments regarding spelling, figure layout etc., which I would appreciate to be addressed by the authors.

Specific comments As stated above, the entire paper is quite technical. Physical oceanography and special characteristics of the region are not really addressed; basically it looks like the North Atlantic and the sub-regions have been selected arbitrarily just to have a reasonable area to investigate which GRACE product fits best. This might be made more clear – this is not so much a paper for readers interested in the North Atlantic, but more for those interested in GRACE and sea level studies.

The authors claim that all three underlying datasets (Argo, GRACE, Altimetry) have been specially processed for this study. While GRACE is introduced in comparatively detail, the introduction of the Argo- and especially the Altimetry data could be more thorough.

There are quite a number of different GRACE products, degrees/orders, Wiener filters etc. Sometimes, it is difficult for the reader to keep track. A shorter, clear designation of the different products, and / or splitting the text into shorter sub-sections might help to make the organization of the paper more clear.

The figures are good, clear and appropriate. However, all explanations (units, variables, etc.) are described in the captions only. It might be an idea to write units directly on the axes or colour bars. In case of multi-panel figures, the panels should be labelled with a), b), c) and so on, or write the name of the GRACE product used in the panel (e.g. ITSG). In the present state, the figures look very clear, but it is tedious to find out

and remember what the blue line in the middle right panel stands for.

P3 L19 How is the alongtrack altimetry data processed? E.g. any de-tiding, smoothing, ...? This could be described in more detail.

P4 L6 This statement is confusing: Why don't the Argo floats resurface in the South Atlantic? Moreover, is this relevant at all for the present study of the North Atlantic?

P4 L16-19: This suggest that the variance-covariance matrices are incorrect for at least one of these time periods. Can you state a reference for this claim?

Table 4: How is - 1.21 (i.e. absolute value larger than 1) for NA with CSR96+A. possible? I would expect values to be in the range -1 ... 1, but not beyond. Moreover, I understand -1 to explain 100% of the variability, but in anticorrelation, suggesting that something is seriously wrong in this particular solution.

Technical corrections

P2 L9: Better write "That study..." – "this study" suggests that this study (submitted to OS) is meant.

Fig 1: The colour bars are too narrow, it is hardly possible to assign colours from the maps to certain numbers. Draw the colour bars larger. Use only a few distinct colour levels (e.g. 0 to 10, 10 to 20, 20 to 30 and so on) both in the map and in the colour bar would make visual assignment much clearer than the present continuous colour scale. Write units (mm, or mm/yr) on the colour bars (see general remarks).

P3 L4 write (Kusche, 2007; Kusche et al., 2009)

Fig. 5: The colour bar is too narrow. See remark to Fig. 1.

P14 L20: "Secondly, compared to Scripps the grids, the other two methods are slightly noisier." This sentence is unclear or grammatically wrong.

Fig. 7: Explain the polygons in the first row in the caption. They should also be

explained in the text; I did not find any explanation except on P17 L10. It remains guesswork to the reader to determine what the three polygons should be. Do I see it correctly that they should be a zonally oriented rectangle, a meridionally oriented rectangle, and a square, all partially overlapping? Did you have a particular reason why you chose this particular location in the North Atlantic? The three polygons could be drawn in different colours or in more different linestyles.

P17 L21: Spelling: "Even though" should be in two words.

Fig. 8: Caption: "Portrait" and "landscape" is ok for paper orientation in printers, but not for describing geographical orientation. Use "meridional" and "zonal" instead.

P19 L8 Polygon H is intoduced only in the next section.

P19 L12 spelling: "tongue"

P20 L1 grammar: delete second "on the": "...chosen based on the criterion that the error does not..."

Fig. 10 caption: "the sum of the two components in blue" – unclear what components are meant. Steric + OBP? The letters "A" to "J" are hard to read, they should be larger and/or in bold face.

P23 L7: "even though" in two words

P23 L9: "overestimation" in one word

P26 L10: grammar: write "They also suffer from..." without "s" at the end of "suffer"

P28 L25: Ablain et al. (2015): The list of authors appears to be incomplete. Similarly for Cabanes et al. (2013), Cazenave et al. (2008), Våge et al. (2009).

---

## Author Comment (AC1) · 5 Sep 2016

Dear reviewer #1,

thank you for the thorough review of the manuscript. We added a rebuttal in the supplement, in which we address all your comments. Additionally, we added an updated version of the manuscript with changes marked.

Best Regards,

Marcel Kleinherenbrink (on behalf of all authors)

Please also note the supplement to this comment:

http://www.ocean-sci-discuss.net/os-2016-50/os-2016-50-AC1-supplement.zip

---

## Referee Comment (RC2) · Anonymous Referee #2 · 3 Oct 2016

General comments

This paper evaluates the sea level budget in regions of the North Atlantic basin using altimetry, GRACE, and steric Argo. Because of the treatment of each dataset is novel and the analysis of budget closure is on sub-basin scales, the subject matter is worthy of publication. However, there are several errors in the description of data processing and in citing previous work that will need to be corrected. More important, the treatment of GIA for both the altimetry and GRACE data is too vague for me to evaluate the claims of sea level budget closure. Another reviewer has already identified several spelling and grammar errors, which I will attempt not to repeat.

The paper needs a better justification for the selection of the North Atlantic as

an area of study. The North Atlantic has been well surveyed before Argo, and other hydrographic profiles or gridded products could have been evaluated. For example, the NOAA Ocean Climate Laboratory's Global total steric sea level anomaly fields include both Argo, XBT, CTD, and other hydrographic data: https://www.nodc.noaa.gov/OC5/3M_HEAT_CONTENT/fsl_global.html

Page 12: I am confused by this statement: "Ultimately, the mean GIA OBP over a polygon (degree 2 and higher, as measured by GRACE) are subtracted from the results, to make them compatible with altimetry." To be compatible with the treatment of the altimetry, a correction based on the same GIA model should have been used. It's not clear from the paper what GIA model was removed from the GRACE data, which is crucial to understanding how the sea level budget was closed for trends. The paper is also unclear about the meaning of the "GIA correction of 10-20%" referred to on page 26. 10 to 20% of what? Since the choice of GIA model used is critical for closure, this issue needs to be more clearly explained.

Minor comments: Introduction, 2nd paragraph. The time periods for a couple of the cited papers is not correct Willis et al., 2008 looked the budget over 2003.5 and 2007.5, not 2003 to 2007. The budget was closed within error by Leuliette and Miller (2009) for 2004-2008, but in Leuliette and Willis (2011) over the period was 2005-2010.5. Page 2, line 4: I'd recommend that throughout the paper the term "GRACE ocean bottom pressure" not be used. Strictly speaking, OBP is the sum of atmospheric and oceanic mass variations, which is what would be recorded by a pressure gauge in the ocean. Here, of course, in equation 1, the authors intend for $H\_OBP$ to reflect the sea level component of ocean mass changes. The IPCC and other authors have opted to call this component "barystatic sea level". I would recommend using this term or "ocean mass component" instead of OBP to avoid confusion. Also, I'd recommend describing how the inverted barometer correction applied to the altimetry data and the GRACE data.

Page 2, line 20: Line 20: Purkey (2014) should be Purkey et al. (2014)

Page 7, line 1: The "inclination weighting" scheme that is a function of the latitude of the measurement and the orbit inclination angle of the satellite and used to generate the University of Colorado time series of GMSL was first suggested by Wang and Rapp [1994] (see the report "Estimation of sea surface dynamic topography, ocean tides, and secular changes from Topex altimeter data" at http://sealevel.colorado.edu/files/pubs/wang_rapp_report_430.pdf). Tai and Wagner (2011) simplified the approach using a spherical Earth approximation. I would recommend citing one or both of these papers. Rather than calling this weighing the "Nerem method," it would more properly be termed the "inclination/latitude weighting" or the Wang and Rapp weighting.

Table 2: EWL is not defined.

---

## Author Comment (AC2) · 12 Oct 2016

Dear reviewer #2,

Thank you for the review of our manuscript. We addressed all your comments in the rebuttal, which is added as a supplement. Additionally, we updated the manuscript with all changes (reviewer #1 and #2) marked.

Best Regards,

Marcel Kleinherenbrink (on behalf of all authors)

Please also note the supplement to this comment:

http://www.ocean-sci-discuss.net/os-2016-50/os-2016-50-AC2-supplement.zip

---

## Author Response (AR1)

**Rebuttal 1**

We would like to thank the anonymous reviewer for the thorough review. We agreed with practically all of the comments and we improved the paper based on these comments. Additionally, we applied some small changes to the methodology section, which are addressed at the bottom of the rebuttal.

The updated manuscript will be added with marked changes.

**Review of Sub-basin scale sea level budgets from satellite altimetry, Argo floats and satellite gravimetry in the North Atlantic by Marcel Kleinherenbrink, Riccardo Riva, and Yu Sun. Submitted to Ocean Science.**

**Reviewed on 19 August 2016.**

**General comments**

**The paper attempts merging sealevel from altimetry with GRACE gravity fields and steric height from Argo floats to close the sea level budget of sub-basin scale areas in the North Atlantic. Different GRACE products are intercompared with convincing results, both in seasonal and interannual variability and trends. The analysis is thorough and worth publication. In some respect the paper appears quite technical, and it should be noted that main aim of the study is to determine which GRACE product and data processing method is most appriate to achieve the desired result, i.e. agreement of observed sealevel (from altimetry), and calculated sealevel from mass changes (GRACE) and steric changes (from Argo floats).**

**The selection of the study area (North Atlantic) and its subdivision into smaller polygons is somewhat arbitrary. I was not able to proofread all equations stated by the authors, I assume that all calculations have been made correctly. I have a few general comments, and a number of minor comments regarding spelling, figure layout etc., which I would appreciate to be addressed by the authors.**

**Specific comments**

**As stated above, the entire paper is quite technical. Physical oceanography and special characteristics of the region are not really addressed; basically it looks like the North Atlantic and the sub-regions have been selected arbitrarily just to have a reasonable area to investigate which GRACE product fits best. This might be made more clear – this is not so much a paper for readers interested in the North Atlantic, but more for those interested in GRACE and sea level studies.**
To make clear that the paper is primarily interesting for sea level and GRACE researchers we changed the title to:
"Sub-basin scale sea level budgets from satellite altimetry, Argo floats and satellite gravimetry: a case study in the North Atlantic"

We added a motivation why we apply the method to the North Atlantic at the end of

the introduction on page 3.
There is no specific requirement for the shape and size of the polygons except for the 1 mm/yr error on the trends, which is mentioned in Sect. 4.3 and Sect. 5.2. However, in the introduction of Sect. 5.2 we state that we try not to cut through the major oceanographic features and the Atlantic is divided in the middle by the Mid-Atlantic ridge.

**The authors claim that all three underlying datasets (Argo, GRACE, Altimetry) have been specially processed for this study. While GRACE is introduced in comparatively detail, the introduction of the Argo- and especially the Altimetry data could be more thorough.**
**+**
**P3 L19: How is the alongtrack altimetry data processed? E.g. any de-tiding, smoothing, ...? This could be described in more detail.**
On page 5/6 we added a short discussion on the geophysical corrections applied in the study. Additionally, we added some references to read more about the geophysical corrections.
Besides this, we believe that the latitude dependent intermission bias of Ablain et al. (2015) and the averaging method are described in enough detail to be reproducible.

We think that the mean steric sea level estimates from Argo are reproducible as well. However, we added a line in the Argo section on page 9 stating that the main purpose of handling Argo this way is to get a full variance-covariance matrix for the steric sea level grid.

**There are quite a number of different GRACE products, degrees/orders, Wiener filters etc. Sometimes, it is difficult for the reader to keep track. A shorter, clear designation of the different products, and / or splitting the text into shorter sub-sections might help to make the organization of the paper more clear.**
We created a table with designations for the various GRACE solutions in section 2. All references to specific GRACE solutions are replace by their respective designations (not marked).

**The figures are good, clear and appropriate. However, all explanations (units, variables, etc.) are described in the captions only. It might be an idea to write units directly on the axes or colour bars. In case of multi-panel figures, the panels should be labelled with a), b), c) and so on, or write the name of the GRACE product used in the panel (e.g. ITSG). In the present state, the figures look very clear, but it is tedious to find out and remember what the blue line in the middle right panel stands for.**
We added units and labels on the axis of all figures and updated captions to all figures. Every colorbar has been widened. For the time series, legends are added.

**P4 L6: This statement is confusing: Why don't the Argo floats resurface in the South Atlantic? Moreover, is this relevant at all for the present study of the North Atlantic?**
What we actually meant was that in 2004 barely any Argo floats were present in the South Atlantic. Sentence removed.

**P4 L16-19: This suggest that the variance-covariance matrices are incorrect for at least one of these time periods. Can you state a reference for this claim?**
That the variance-covariance matrices vary not much month-to-month over these

periods is stated on the README of their website.
ftp://ftp.csr.utexas.edu/outgoing/grace/README

We added a reference to Klinger 2016. They showed that the variability of the gravity fields over the ocean during repeat orbits is much larger and therefore the variance-covariance matrices should be quite different during these periods.

**Table 4: How is - 1.21 (i.e. absolute value larger than 1) for NA with CSR96+A. possible? I would expect values to be in the range -1 ... 1, but not beyond. Moreover, I understand -1 to explain 100% of the variability, but in anticorrelation, suggesting that something is seriously wrong in this particular solution.**
There is a difference between computing the explained variance between two time series and between a model fitted through the time series with the original time series. In case a model is explaining a fraction of the variance it is indeed the case that the values will never exceed -1 or 1. However we are comparing two independent time series, suppose 'A' and 'B'. Let's recall the formula for explained variance (EV):

EV=1-var(A-B)/var(A)

Suppose now that B has a variance 10 times larger than A, then var(A-B) will become big, while var(A) is still small. As a result var(A-B)/var(A) can become larger than 2 and therefore the EV more negative than -1. B is in our case Argo+GRACE and A altimetry.

We added the sentence:
'Note that the value -1.21 for the CSR96-W gravity fields indicates that variance increases after its subtraction from altimetry, which indicates that the Argo+GRACE time series is substantially noisier than the altimetry time series.'

**Technical corrections**

**P2 L9: Better write "That study..." – "this study" suggests that this study (submitted to OS) is meant.**
Changed 'this' into 'that'.

**Fig 1: The colour bars are too narrow, it is hardly possible to assign colours from the maps to certain numbers. Draw the colour bars larger. Use only a few distinct colour levels (e.g. 0 to 10, 10 to 20, 20 to 30 and so on) both in the map and in the colour bar would make visual assignment much clearer than the present continuous colour scale. Write units (mm, or mm/yr) on the colour bars (see general remarks).**
We changed the color levels, increase the width of the colorbar and added units.

**P3 L4: write (Kusche, 2007; Kusche et al., 2009)**
Updated.

**Fig. 5: The colour bar is too narrow. See remark to Fig. 1.**
The colorbars have been widened.

**P14 L20: "Secondly, compared to Scripps the grids, the other two methods are slightly noisier." This sentence is unclear or grammatically wrong.**
Changed to: "Secondly, the grid computed in this study and the Glorys grid exhibit

more short-wavelength spatial variability than the Scripps grid."

**Fig. 7: Explain the polygons in the first row in the caption. They should also be explained in the text; I did not find any explanation except on P17 L10. It remains guesswork to the reader to determine what the three polygons should be. Do I see it correctly that they should be a zonally oriented rectangle, a meridionally oriented rectangle, and a square, all partially overlapping? Did you have a particular reason why you chose this particular location in the North Atlantic? The three polygons could be drawn in different colours or in more different linestyles.**
In the caption of Fig. 6 the polygons are introduced. On page 16 a sentence is added: "The location is chosen in the middle of the Atlantic to avoid effects of hydrological leakage." The lines of the polygons have now different colors to make them more distinguishable.

**P17 L21: Spelling: "Even though" should be in two words.**
Spelling changed.

**Fig. 8: Caption: "Portrait" and "landscape" is ok for paper orientation in printers, but not for describing geographical orientation. Use "meridional" and "zonal" instead.**
Changed to "meridional" and "zonal".

**P19 L8: Polygon H is intoduced only in the next section.**
Changed to "west of the Mediterranean".

**P19 L12: spelling: "tongue"**
Spelling changed.

**P20 L1: grammar: delete second "on the": "...chosen based on the criterion that the error does not..."**
Second "on the" removed.

**Fig. 10: caption: "the sum of the two components in blue" – unclear what components are meant. Steric + OBP? The letters "A" to "J" are hard to read, they should be larger and/or in bold face.**
Fig. 8: The letters are in bold font.
Fig. 9: Caption updated "the sum of steric sea level and OBP in blue"
Fig. 10: Idem.

**P23 L7: "even though" in two words**
Spelling changed.

**P23 L9: "overestimation" in one word**
Spelling changed.

**P26 L10: grammar: write "They also suffer from..." without "s" at the end of "suffer"**
The 's' removed

**P28 L25: Ablain et al. (2015): The list of authors appears to be incomplete. Similarly for Cabanes et al. (2013), Cazenave et al. (2008), Våge et al. (2009).**

In the first submission references with more than six authors were made in the following way:
author 1, author 2, author 3, author 4, author 5, …., last author, (year). Article. Journal, etc.

We added now all authors for the following six references with more than six authors:
Ablain et al. (2015)
Cabanes et al. (2013)
Cazenave et al. (2008)
Von Schuckmann et al. (2014)
Storto et al. (2015)
Vage et al. (2009)

**Additional corrections by author**

**P10 L5 and L12:** Changed 'correlations' into 'covariances'
**P10 L8 and Eq. 21:** Changed steric sea level 's' into 'h'
**P10 L10:** Changed the subscript 'ssla' to 'ssl,g'
**P10 L13:** Removed 'g' from subscripts
**P5 L25:** Changed 'with calibrated variance-covariance matrices' into 'with full variance-covariance and normal matrices'

**Rebuttal 2**

We would like to thank reviewer #2 for his review. We agreed with most of the comment and implemented the changes as requested.

The updated manuscript will be added with marked changes.

**General comments**

**This paper evaluates the sea level budget in regions of the North Atlantic basin using altimetry, GRACE, and steric Argo. Because of the treatment of each dataset is novel and the analysis of budget closure is on sub-basin scales, the subject matter is worthy of publication. However, there are several errors in the description of data processing and in citing previous work that will need to be corrected. (1) More important, the treatment of GIA for both the altimetry and GRACE data is too vague for me to evaluate the claims of sea level budget closure. Another reviewer has already identified several spelling and grammar errors, which I will attempt not to repeat. (2) The paper needs a better justification for the selection of the North Atlantic as an area of study. The North Atlantic has been well surveyed before Argo, and other hydrographic profiles or gridded products could have been evaluated. For example, the NOAA Ocean Climate Laboratory's Global total steric sea level anomaly fields include both Argo, XBT, CTD, and other hydrographic data: https://www.nodc.noaa.gov/OC5/3M_HEAT_CONTENT/fsl_global.html**

(1) The description of the GIA correction is extended and clarified. This is further addressed below at 'page 12' and 'page 26'.

(2) Reviewer #1 also brought this to our attention. We decided to update the title to: "Sub-basin scale sea level budgets from satellite altimetry, Argo floats and satellite gravimetry: a case study in the North Atlantic". Secondly, we added a motivation why apply the method to the North Atlantic in the introduction. We needed a location where enough Argo observations were available during the period 2004-2014. Additionally, the North Atlantic is a quite dynamical region, with the Gulf Stream, a neighbouring ice sheet and large gradients in GIA, so it was possible to investigate the method over regions with varying conditions.

**Page 12: I am confused by this statement: "Ultimately, the mean GIA OBP over a polygon (degree 2 and higher, as measured by GRACE) are subtracted from the results, to make them compatible with altimetry." To be compatible with the treatment of the altimetry, a correction based on the same GIA model should have been used. It's not clear from the paper what GIA model was removed from the GRACE data, which is crucial to understanding how the sea level budget was closed for trends.**

For altimetry and GRACE we consistently use the solution of Peltier et al. (2015). For altimetry it is corrected for the geoid response and for GRACE the EWL response.

We adjusted the text on page 9 (under altimetry processing), to clarify that for altimetry and GRACE the same model is used. On page 12 we clarified and extended the text related to the correction for GRACE.

**Page 26: The paper is also unclear about the meaning of the "GIA correction of 10-20%" referred to on page 26. 10 to 20% of what? Since the choice of GIA model used is critical for closure, this issue needs to be more**

**clearly explained.**
The sentence is updated to "an error of 10-20 % on the GIA correction is assumed".

**Minor comments**

**Introduction, 2nd paragraph. The time periods for a couple of the cited papers is not correct Willis et al., 2008 looked the budget over 2003.5 and 2007.5, not 2003 to 2007. The budget was closed within error by Leuliette and Miller (2009) for 2004-2008, but in Leuliette and Willis (2011) over the period was 2005-2010.5.**
The periods have been corrected.

**Page 2, line 4: I'd recommend that throughout the paper the term "GRACE ocean bottom pressure" not be used. Strictly speaking, OBP is the sum of atmospheric and oceanic mass variations, which is what would be recorded by a pressure gauge in the ocean. Here, of course, in equation 1, the authors intend for H_OBP to reflect the sea level component of ocean mass changes. The IPCC and other authors have opted to call this component "barystatic sea level". I would recommend using this term or "ocean mass component" instead of OBP to avoid confusion. Also, I'd recommend describing how the inverted barometer correction applied to the altimetry data and the GRACE data.**
The IB correction is applied as a linear function of regional sea level pressure variations with respect to the time-varying mean atmospheric sea level pressure over the oceans. Fluctuations in atmospheric pressure are caused by mass fluctuations in the atmosphere.
Altimetry without IB correction measures steric changes + ocean mass changes. However, by applying the IB correction regional variations of atmospheric mass (with respect to the global mean over the oceans) are also taken into account.
Since the mean atmospheric mass over the oceans is not part of our measurements we agree with the reviewer that the term 'ocean bottom pressure' is not appropriate. However, since regional variations of atmospheric mass/pressure (by the IB correction) are taken into account, we think the term 'ocean mass' is also not appropriate.
According to Gregory et al. (2012): "The barystatic effect on sea level change is the mass of freshwater added or removed, converted to a volume using a reference density of 1000 kg m^{-3}." As it does not take into account the IB correction, 'barystatic sea level' is also not appropriate.
As a solution, we will now refer to 'Mass Component (MC)' or 'mass' as the IB corrected ocean mass. Throughout the whole manuscript including Eq. (1), we now use the term MC. We make clear that the MSL and MC in Eq. (1) are inverse barometer corrected.

Furthermore, a short description is included for all the corrections applied to the altimetry, including the inverted barometer (IB) correction, which is part of the dynamic atmosphere correction. We slightly adjusted the description of the GRACE data, to clarify that time-varying global mean atmospheric mass over the ocean is excluded, as it is also for altimetry.

**Page 2, line 20: Purkey (2014) should be Purkey et al. (2014).**
Changed to Purkey et al. (2014).

**Page 7, line 1: The "inclination weighting" scheme that is a function of the latitude of the measurement and the orbit inclination angle of the satellite and used to generate the University of Colorado time series of GMSL was**

**first suggested by Wang and Rapp [1994] (see the report "Estimation of sea surface dynamic topography, ocean tides, and secular changes from Topex altimeter data" at http://sealevel.colorado.edu/files/pubs/wang_rapp_report_430.pdf). Tai and Wagner (2011) simplified the approach using a spherical Earth approximation. I would recommend citing one or both of these papers. Rather than calling this weighing the "Nerem method," it would more properly be termed the "inclination/latitude weighting" or the Wang and Rapp weighting.**

We agree with the reviewer that the derivation of the inclination weighting technique was described by Wang and Rapp (1994). Therefore we changed the name of the weighting method to Wang and Rapp weighting. Additionally, both Tai and Wagner (2011) and Wang and Rapp (1994) references are included.

**Table 2: EWL is not defined.**
We changed EWL into EWH to be consistent with the rest of the manuscript. We define EWH again in table 4.

**Additional changes by author**

Page 9, line 6: Changed 'the EWHs' into 'MC from GRACE'.
Page 13, line 12: Added a reference to Tamisiea (2011).

We noted that sea level anomalies in Eq. (1) were not exactly what is computed in methodology. First of all, for all time series the mean is removed, because they have different reference periods. So, at the end of Sect. 3.1, 3.2 and 3.3 we mention that the mean of the msl, steric and mass time series is removed. Secondly, we clearly state in Eq. (1) that the mean sea level and the mass component are GIA-corrected.

**References**

Gregory et al. (2013). Twentieth-century global-mean sea level rise: Is the whole greater than the sum of the parts?. *Journal of Climate*, *26*(13), 4476-4499.

[revised manuscript text omitted]

---

## Author Response (AR2)

Dear Dr. Hoppema,

Thanks for the thorough review of our revised manuscript. We managed to address all of your comments. The details are listed below.

Best Regards,

Marcel Kleinherenbrink (on behalf of all authors)

**Dear Dr. Kleinherenbrink and co-authors,**

**Thank you for your revised manuscript.**
**I have listed some last comments to your revised manuscript below. Provided these are accounted for, the manuscript will be accepted for publication in Ocean Science.**

**Thanks and best wishes**
**Mario Hoppema**

**Non-public comments to the Author:**

**List with comments:**

**There are so many abbreviations in this manuscript that a table with all of them and their meaning or use is highly appropriate. Please add it.**
A table with abbreviations is added in Appendix B.

**Title: I suggest to add "Ocean" after Atlantic**
**… a case study in the North Atlantic Ocean**
Title is updated.

**First sentence of abstract: "In this study for the first time an attempt is made to close the sea level budget on a sub-basin scale in terms of trend, annual amplitude and residual time series, after removing the trend, the semi-annual and annual signals." I find it strange to read that an attempt is made to close the sea level budget on a sub-basin scale in terms of trend, but then later in the sentence that the trend will be removed. Is this correct, or could it be phrased differently?**
Changed to: In this study for the first time an attempt is made to close the sea level budget on a sub-basin scale in terms of trend and amplitude of the annual cycle. We also compare the residual time series after removing the trend, the semi-annual and the annual signals.

**P1 L9 define DKK5 and CSR**
The abbreviation DDK is not even defined in the original paper, where the filter is first described (Kusche, 2007) and the paper where the term DDK is first used (Kusche et al., 2009).  We contacted the author of both papers, Prof. Kusche. He told us that the second "D" and the "K" stand for "Decorrelation Kernel", but he did not know the meaning of the first "D". We also emailed Susanne Werth, one of the former students of Prof. Kusche, who is supposed to know the abbreviation, but she did not respond yet.

Based on the anisotropic properties of the filter, our best guess is that the first "D" stands for "Directional", we added this to the manuscript. We assume that Susanne Werth will

reply within a couple of weeks, so we are able to update the definition during proof reading if necessary. Otherwise, we will use the term "Decorrelation filter (DDK)", which is used commonly  in other publications.

Directional Decorrelation Kernel (DDK) and Center for Space Research (CSR) are defined in the abstract once and also in the main text. CSR is defined at line 4 of page 4 in the main text and DDK on line 5 of page 5.

**P1 L10 anisotropic (typo)**
Updated.

**P1 L10 „d/o" I am not familiar with this notation and I can imagine more readers will not be. Please explain or use different notation.**
Changed to 'degree and order'.

**P1 L13 define ITSG**
Institute for Theoretical geodesy and Satellite Geodesy (ITSG) is define in the abstract once and once in the main text on page 5, line 7.

**P1 L16 Why not just: In seven of ten …**
Line 14 and 17: changed to "nine of ten" and "seven of ten".
Also changed in the results and conclusion sections.

**P1, L17 lacks**
**P1, L18 suffers**
Instead, we changed "solution" to "solutions".

**P1 L19 … the semi-annual and the annual signals …**
Updated.

**P1 L22 North Atlantic Ocean**
Updated.

**Not being a specialist in this field, I am wondering about the use of the term "sea level budget" here. Maybe it would be useful to the reader to explain in one or two sentences what a sea level budget is the way you are using the term.**
Added the following sentences: "If the sum of individual components is statistically equal to the total sea level variations the budget is closed. Total sea level variations and its components are observed by in-situ and satellite measurements, but can also be modelled."

**P2 L2 Several studies have attempted …**
Updated.

**P2 L10 … between the middle of the years 2003 and 2007 (i.e. 2003.5 and 2007.5) …**
Updated.

**P2 L10-11 … variability; however, the 4-year trends did not agree. (punctuation)**
Updated.

**P2 L15 aforementioned (not: beforementioned)**
Updated.

**P2 L26-27 delete: (Von Schuckmann et al., 2014) (was referred to earlier in that sentence)**
Reference removed.

**P2 L27 30-60°N (format)**
Reformatted.

**P2 L30-31 It looks better like this: … the ECCO (Estimating the Circulation and Climate of the Ocean) model.**
Updated.

**P3 L1-2 "Secondly, we address the effect of several processing steps of particularly on gravimetry data in terms of trend …" Something is wrong with this sentence, second "of"? Please correct.**
Changed sentence to: Secondly, we address the effect of several processing steps particularly on gravimetry data in terms of trend, annual amplitude and (residual) time series.

**P4 L1 time-variable (hyphen)**
Updated.

**P4 L4-5 "In this study the release 5 monthly spherical harmonic solutions computed at CSR (Tapley et al., 2004) up to degree and order 60 and 96 are used …" This sentence is very hard to read and understand. Please correct it.**
The sentence is split in two, so that: In this study the release 5 monthly spherical harmonic solutions computed at the Center for Space Research (CSR) (Tapley et al., 2004), together with the ITSG-GRACE2016 solutions (Klinger et al., 2016) computed at the Institute for Theoretical geodesy and Satellite Geodesy (ITSG). The CSR solutions are computed up to degree and order 60 and 96, while the ITSG solutions are computed up to degree and order 90.

**P5 L31 (same as in the abstract) „d/o" I am not familiar with this notation and I can imagine more readers will not be. Please explain or use different notation.**
Changed to: degree and order.

**P6 caption of Table 2: List of geophysical corrections applied … (+s)**
Updated the sentence to: List of geophysical corrections applied in this study and for the MSLs of NOAA.

**P6 L6 corrections**
Updated.

**P6 L7 35-second (hyphen)**
Updated.

**P6 L22 more prone (not: proner)**
Changed.

**P7 L8 geophysical (typo)**
Updated.

**P7 L8-9 "At halfway the North Atlantic" is very unspecific. Isn't there a way to be more specific?**
Changed to: In the middle of the North Atlantic Ocean (approximately 40°N).

**P9 L24 temperature/salinity (no capitals)**
Updated.

**P9 L28 The right term is "potential temperature". (You may want to explain that this is conservative)**
In TEOS-10 there is a difference between potential temperature and conservative temperature. The manual provides the following: "The Gibbs function approach allows the calculation of internal energy, entropy, enthalpy, potential enthalpy and the chemical potentials of seawater as well as the freezing temperature, and the latent heats of melting and of evaporation. These quantities were not available from the International Equation of State 1980 but are essential for the accurate accounting of "heat" in the ocean and for the consistent and accurate treatment of air-sea and ice-sea heat fluxes. For example, the new TEOS-10 temperature variable, Conservative Temperature, $\Theta$, is defined to be proportional to potential enthalpy and is a very accurate measure of the "heat" content per unit mass of seawater; $\Theta$ is two orders of magnitude more conservative than potential temperature $\theta$."

Therefore, we changed the sentence to:
"to conservative temperature $\Theta$ as defined in the TEOS-10 user manual (IAPSO, 2010)."

**P10 L1 dbar (no capital for bar)**
Updated.

**P10 L1 1000 m**
Updated.

**P10 L2 m s-2 (format)**
Updated.

**P10 L3 1000 m (also at several other places: use m for meter)**
Updated everywhere.

**P10 L11 longitude-latitude (write full)**
Written fully.

**P11 L21 move "respectively" to after "signals"**
Respectively moved.

**P12 L1 … in Appendix A.**
Updated.

**P12 L5 effect (not: affect)**
Updated.

**P13 L24-25 north of 30°N (instead of: above 30 degrees latitude)**
Updated.

**P14 Fig 4 caption correct to: Differences in sea level trends computed with and without a latitude-dependent intermission bias**
Updated.

**P16 first sentence: "In Fig. 6 the trends and amplitudes of the CSR96-DDK compared with those obtained from CSR60-, CSR96- and ITSG90-W." This sentence is not grammatical. Something is missing here. Please correct.**
Sentence updated to: "In Fig. 6 the trends and amplitudes of the CSR96-DDK solution are compared with those obtained from CSR60-, CSR96- and ITSG90-W."

**P18 caption Fig. 6. Move "respectively" to the last position of the sentence.**
Respectively moved.

**P18 L24 north of 60°N (instead of: above degree 60)**
Changed to: "towards 60°N."

**P20 L12-13 … just to the west of Africa …**
Updated.

**P20 L19-20 … are found south of 35°N and negative trends north of it, with …**
Updated.

**P20 L24 … will be discussed below.**
Updated.

**L26 The North Atlantic Ocean is …**
Updated.

**P20 L28-29 "…on the criterion that the error on the does not exceed 1 mm yr-1." Sentence is not grammatical. Something is missing or over. Please correct.**
Changed sentence to: "Just as in Sect. 4.3, the size of the regions is chosen such that the error on the trends does not exceed 1 mm yr$^{-1}$.

**L3 "…but the others do not." I am not sure what you mean here. Is it: … but the others are not."? Still not sure what you want to say with this.**
Changed to: "whereas the other solutions do not close the budget."

**P24 caption Table 4. … for the complete North Atlantic from 0°-65°N.**
Updated.

**P24 caption Table 4. This part is not clear at all. Please correct and clarify: " *GIA Absolute Sea Level (ASL) correction subtracted from altimetry MSL and GIA Equivalent Water Height (EWH) correction subtracted from the GRACE MC." (for example, the star does not occur in the table)**
Changed to: GIA Absolute Sea Level (ASL*) correction subtracted from altimetry MSL and GIA Equivalent Water Height (EWH**) correction subtracted from the GRACE MC.

Included stars in the corresponding columns.

**P24 L11 Move "respectively" to the end of the sentence.**
Respectively moved.

**P26 L4 Atlantic Ocean**
Changed to: "North Atlantic Ocean".

**P26 L5 east coast (no capitals)**
Capitals removed.

**P26 L10 east (no capital)**
Capital removed.

**P26 L33 anisotropic (typo)**
Updated.

**P27 L5 Mediterranean Sea**
Updated.

**P27 L6 Atlantic Ocean**
Updated.

**References are not listed alphabetically. Please do so. Also please use the Ocean Science format.**
References are alphabetically listed and "Ocean Science" format used.

**Deep-Sea Research (with hyphen)**
Updated.

**Additional changes by author:**

Page 5, line 23: Defined TEOS-10. Thermodynamic Equation Of Seawater-2010 (TEOS-10).

Page 6, line 20: Defined NOAA. National Oceanic and Atmospheric Administration (NOAA).

Page 17, line 24: Changed "grid cells approximately" to "grid cells of approximately"

Several pages:  Changed "North Atlantic" to "North Atlantic Ocean".

[revised manuscript text omitted]